# Complex plumages spur rapid color diversification in kingfishers (Aves: Alcedinidae)

**Chad M Eliason[1,2]\*, Jenna M McCullough[3], Shannon J Hackett[2], Michael J Andersen[3]**

[1]Grainger Bioinformatics Center, Field Museum of Natural History, Chicago, United States; [2]Negaunee Integrative Research Center, Field Museum of Natural History, Chicago, United States; [3]Department of Biology and Museum of Southwestern Biology, University of New Mexico, Albuquerque, United States

**Abstract** Colorful signals in nature provide some of the most stunning examples of rapid phenotypic evolution. Yet, studying color pattern evolution has been historically difficult owing to differences in perceptual ability of humans and analytical challenges with studying how complex color patterns evolve. Island systems provide a natural laboratory for testing hypotheses about the direction and magnitude of phenotypic change. A recent study found that plumage colors of island species are darker and less complex than continental species. Whether such shifts in plumage complexity are associated with increased rates of color evolution remains unknown. Here, we use geometric morphometric techniques to test the hypothesis that plumage complexity and insularity interact to influence color diversity in a species-rich clade of colorful birds—kingfishers (Aves: Alcedinidae). In particular, we test three predictions: (1) plumage complexity enhances interspecific rates of color evolution, (2) plumage complexity is lower on islands, and (3) rates of plumage color evolution are higher on islands. Our results show that more complex plumages result in more diverse colors among species and that island species have higher rates of color evolution. Importantly, we found that island species did not have more complex plumages than their continental relatives. Thus, complexity may be a key innovation that facilitates evolutionary response of individual color patches to distinct selection pressures on islands, rather than being a direct target of selection itself. This study demonstrates how a truly multivariate treatment of color data can reveal evolutionary patterns that might otherwise go unnoticed.

**\*For correspondence:** celiason@fieldmuseum.org

## Editor's evaluation

This important work advances our understanding of the factors that affect the speed of colour evolution in birds and the resulting diversification patterns. It provides compelling evidence that more complex plumage coloration can lead to rapid colour evolution in kingfishers, and will pave the way for more comprehensive analyses that fully embrace the multidimensional nature of colour variation. Hence, the results will be of broad interest to ornithologists and evolutionary biologists in general.

## Introduction

Understanding spatial and temporal trends in phenotypic diversity continues to be an important challenge in evolutionary biology. Colorful signals in birds are a good case study for a rapidly evolving phenotype that shows variation at broad spatial (*Cooney et al., 2022*) and phylogenetic scales (*Cooney et al., 2019*). Birds produce colorful plumage patterns with a combination of two mechanisms: light

**eLife digest** Birds are among the most colorful animals on Earth. The different patterns and colors displayed on their feathers help them to identify their own species, attract mates or hide from predators.

The bright plumages of birds are achieved through either pigments (such as reds and yellows) or structures (such as blues, greens or ultraviolet) inside feathers, or through a combination of both pigments and structures. Variation in the diversity of color patterns over time can give a helpful insight into the rate of evolution of a species. For example, structural colors evolve more quickly than pigment-based ones and can therefore be a key feature involved in species recognition or mate attraction.

Studying the evolution of plumage patterns has been challenging due to differences in the vision of humans and birds. However, recent advances in technology have enabled researchers to map the exact wavelengths of the colors that make up the patterns, allowing for rigorous comparison of plumage color patterns across different individuals and species.

To gain a greater understanding of how plumage color patterns evolve in birds, Eliason et al. studied kingfishers, a group of birds known for their complex and variable color patterns, and their worldwide distribution. The experiments analyzed the plumage color patterns of 72 kingfisher species (142 individual museum specimens) from both mainland and island populations by quantifying the amount of different wavelengths of light reflecting from a feather and accounting for relationships among species and among feather patches.

The analyzes showed that having more complex patterns leads to a greater accumulation of plumage colors over time, supporting the idea that complex plumages provide more traits for natural or sexual selection to act upon.

Moreover, in upper parts of the bodies, such as the back, the plumage varied more across the different species and evolved faster than in ventral parts, such as the belly or throat. This indicates that sexual selection may be the evolutionary force driving variation in more visible areas, such as the back, while patterns in the ventral part of the body are more important for kin recognition.

Eliason et al. further found no differences in plumage complexity between kingfishers located in island or mainland habitats, suggesting that the isolation of the island and the different selection pressures this may bring does not impact the complexity of color patterns. However, kingfisher species located on islands did display higher rates of color evolution. This indicates that, regardless of the complexity of the plumage, island-specific pressures are driving rapid color diversification.

Using a new multivariate approach, Eliason et al. have unearthed a pattern in plumage complexity that may otherwise have been missed and, for the first time, have linked differences in color pattern on individual birds with evolutionary differences across species. In doing so, they have provided a framework for future studies of color evolution. The next steps in this research would be to better understand why the island species are evolving more rapidly even though they do not have more complex plumage patterns and how the observed color differences relate to rapid rates of speciation.

absorption by pigments and light scattering by feather nanostructures (*Shawkey and D'Alba, 2017*). Whereas melanin- and carotenoid-based coloration are produced by chemical pigments and absorb light waves, structural colors are produced by the physical interaction of light waves and nanometer-scale variations in the feather integument (*Prum, 2006*). Within birds, structural colors produce a wide array of color, including blue-green colors, glossy blacks, and iridescence. Because structural colors are more evolutionarily labile than pigment-based colors, they have faster evolutionary rates (*Eliason et al., 2015*) and are considered key innovations in some clades (e.g., African starlings; see *Maia et al., 2013b*). In addition to how color is produced, birds also vary in where they deploy colors in their plumage (*Stoddard and Prum, 2011*). Yet, studying color pattern evolution has been historically difficult due to our inability to perceive UV color (*Eaton, 2005*) and challenges with quantifying and analyzing complex color patterns (*Mason and Bowie, 2020*).

While the color of individual patches can be influenced in different directions by multiple selective factors (*Cuthill et al., 2017*), the deployment of color in distinct patterns appears to be constrained developmentally (*Hidalgo et al., 2022*). Since selection can only act on existing variability, such as

distinct plumage patches across a bird's body, ancestrally shared developmental bases of plumage patterns might act as a brake on color evolution (*Price and Pavelka, 1996*; *Hidalgo et al., 2022*; but see *Felice et al., 2018*). For example, in a hypothetical, uniformly colored species with strong developmental constraints that limit independent variation in color among patches, selection on the color of any single patch would cause the whole plumage to change in tandem. By contrast, if a species is variably colored (i.e., patchy, and therefore has a more complex plumage) with few constraints on the direction of color variation for different patches, selection can act on different aspects of coloration (*Brooks and Couldridge, 1999*). On a macroevolutionary scale, we would predict greater color divergence in a clade with an ancestrally complex plumage pattern because there is more standing color variation among patches upon which selection can act. On a more microevolutionary scale, however, intraspecific plumage complexity (i.e., the degree of variably colored patches across a bird's body) could be a key innovation that drives rates of color evolution in birds and should be considered alongside ecological and geographic hypotheses.

Islands have been considered natural laboratories for studying evolution because they often lack natural predators and competitors due to their geographic isolation (*Losos and Ricklefs, 2009*). Compared to life history (*Covas, 2012*; *Losos et al., 1998*; *Novosolov et al., 2013*), behavior (*Buglione et al., 2019*; *Roff, 1994*), and morphological traits (*Clegg and Owens, 2002*; *Wright et al., 2016*), signals used in mating and social contexts have been less commonly explored in the context of island evolution. Yet, previous work has shown increased color polymorphism in island snails (*Bellido et al., 2002*; *Ożgo, 2011*) and lizards (*Corl et al., 2010*). Within birds, island species tend to be less sexually dimorphic and have simpler songs (*Price, 2008*). Island birds have also been shown to have darker colors and simpler plumage patterns (*Bliard et al., 2020*; *Doutrelant et al., 2016*). Under a species-recognition hypothesis, these shifts are thought to be driven by reduced competition on islands (*Martin et al., 2015*; *Doutrelant et al., 2016*), as fewer competitors would lower the risk of hybridization and cause a reduction in signal distinctiveness on islands (*West-Eberhard, 1983*). Despite these advances, we lack a detailed understanding of color evolution within, rather than between, island and mainland clades. For example, are changes in plumage color on islands also accompanied by bursts in phenotypic evolutionary rates, as has been shown for morphological traits in other groups (*Millien, 2006*; *Thomas et al., 2009*; *Woods et al., 2020*)?

Two hallmarks of kingfishers (Aves: Alcedinidae) are their complex plumage patterns (*Eliason et al., 2019*) and their island distributions (*Andersen et al., 2018*; *McCullough et al., 2019*). Kingfishers encompass a wide variety of colors—from the aquamarine-colored back of the common kingfisher (*Alcedo atthis*) to the brilliant silver back of the southern silvery-kingfisher (*Ceyx argentatus*), as well as the purple rump of the ultramarine kingfisher (*Todiramphus leucopygius*). They also run the gamut of plumage complexity, including intricate scalloped plumage of the spotted kingfisher (*Actenoides lindsayi*) and the contrasting hues of the black-backed dwarf-kingfisher (*Ceyx erithaca*). The family is widely distributed across the globe, but their center of diversity is the Indo-Pacific, including island clades in Wallacea and Melanesia that have recently been highlighted for their high diversification rates (*Andersen et al., 2018*). These same island clades, specifically within the woodland kingfisher genus *Todiramphus* and *Ceyx* pygmy-kingfishers, also have elevated color diversity (*Eliason et al., 2019*) and complex geographic histories. These genera include many allopatric, island-endemic taxa, as well as harboring a high degree of sympatry on islands (*Andersen et al., 2015*). For example, there are 10 species of kingfishers that occur on the Indonesian island Halmahera, 5 of which are in the genus *Todiramphus*. There are also multiple instances of sympatry in *Ceyx*, including on New Guinea, the Philippines, and the Solomon Islands (*Andersen et al., 2013*). Smaller population sizes, isolation, and genetic drift could potentially explain high rates of color evolution in island kingfishers, making them an ideal system to investigate the interplay between key innovations (complex plumages) and geographic isolation (i.e., spatial opportunity) in driving rapid color evolution.

In this study, we implement geometric morphometric techniques to investigate complex plumage pattern evolution across kingfishers. We hypothesized that potential constraints limiting where and how color is produced on a bird's body should also limit evolutionary changes between species. Specifically, if complex plumages are a key innovation enabling rapid rates of color evolution (Prediction 1), and if plumage complexity is lower on islands (Prediction 2), then insularity and plumage complexity should both influence the direction and rate of change of plumage coloration (Prediction 3; see *Figure 1B*). We tested these predictions using UV–vis reflectance spectrophotometry of

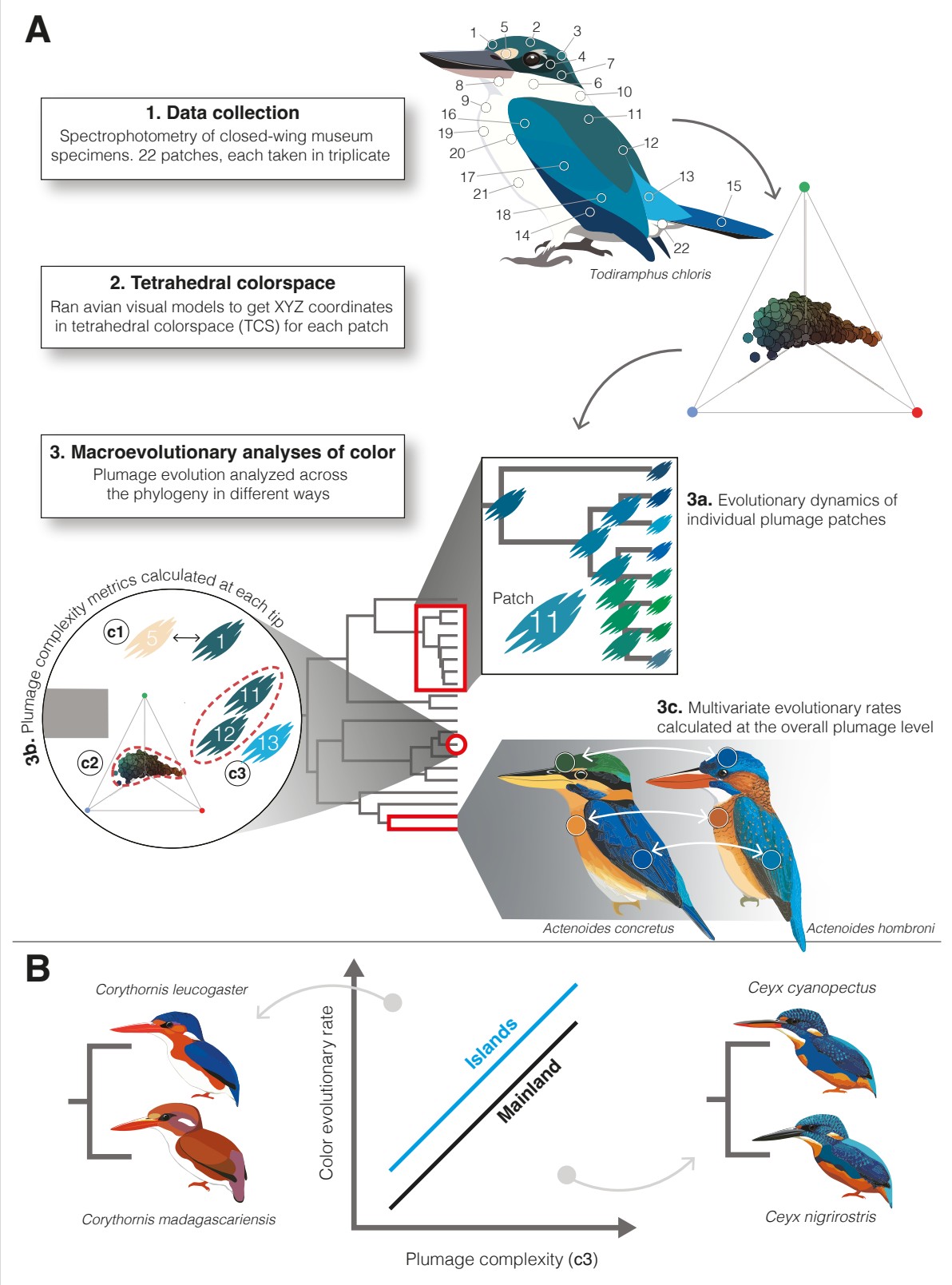

**Figure 1.** Illustrative guide to methods used to study kingfisher plumage coloration. (**A**) Flow chart depicting our process: (1) spectrophotometry of 22 plumage patches on closed-wing museum specimens, (2) conversion of data to tetrahedral colorspace coordinates, and (3) different ways we analyzed these data across the kingfisher phylogeny. We analyzed how individual patch colors evolved using multivariate comparative methods (3a). To estimate complexity at the intraspecific level (3b), we calculated three different metrics for each tip in the phylogeny: average pairwise distance among color

*Figure 1 continued on next page*

*Figure 1 continued*

patches (metric c1); the color volume (i.e., range) of all plumage patches in colorspace (metric c2); and the number of contiguous color patches that would be perceived as the same color by a bird (metric c3). We calculated interspecific rates of overall plumage color evolution using multivariate rate tests (3c). (**B**) We predicted faster rates of color evolution on islands (blue line) and in species with more complex plumages. Yet, there are examples of cases in which this relationship may be reversed (e.g., see insets showing species pairs with simple plumages and diverse colors, left, as well as complex plumages and similar colors, right). Illustrations created by Jenna McCullough.

The online version of this article includes the following figure supplement(s) for figure 1:

**Figure supplement 1.** Complexity metrics are correlated.

**Figure supplement 2.** Plumage patches measured in kingfishers.

museum specimens and multivariate comparative methods. Our study of the interplay between the arrangement of color patches, interspecific competition, and geography sheds light more broadly on the role of spatial opportunity in phenotypic evolution.

## Results
### Holistic assessment of plumage color variation

Plumage coloration is highly multivariate, varying both within feathers, among feather regions on a bird, between sexes, and among species. To visualize trends in these data, we conducted a partitioning of variance analysis that revealed two distinct modes of color variation within kingfishers: (1)

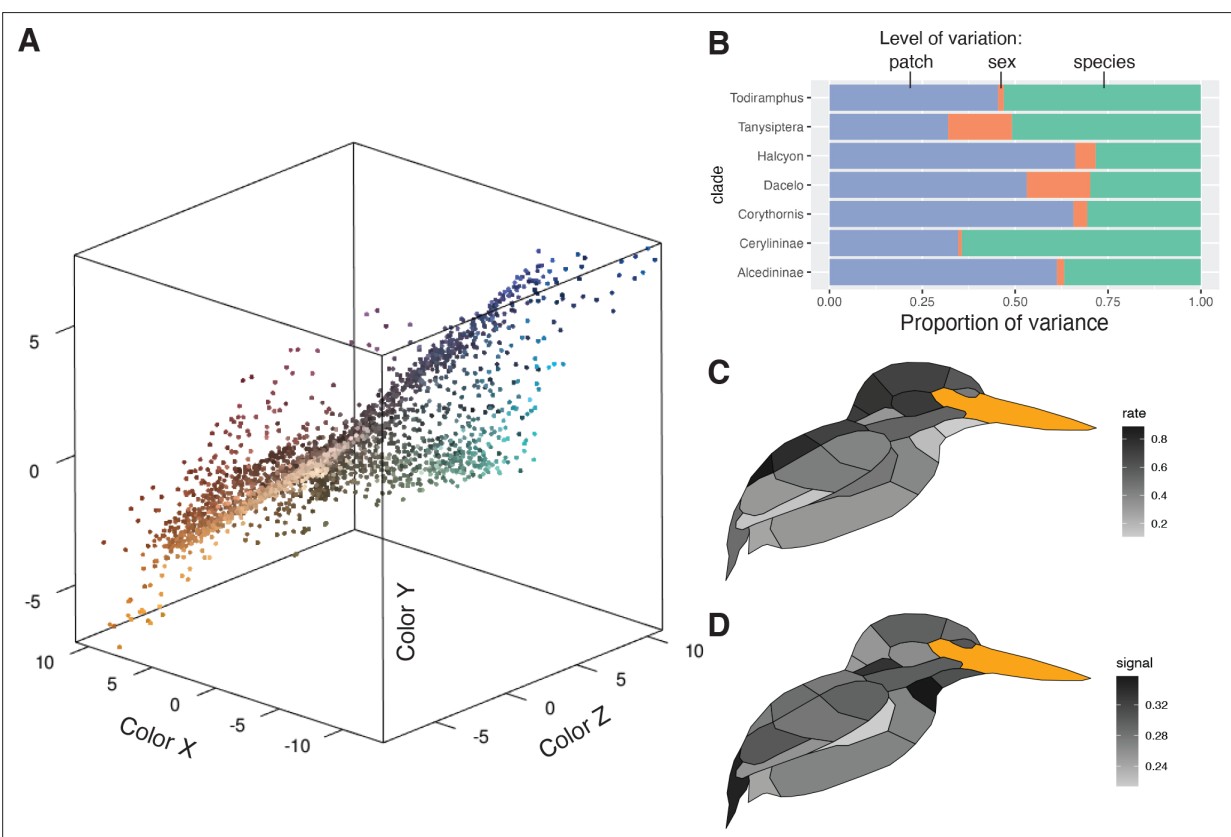

**Figure 2.** Perceptually uniform colorspace and color variation in kingfishers. (**A**) Color data, with points being the average of three plumage patch measurements for each individual (*N* = 3101). Colors are estimated from a human visual system using spec2rgb in pavo (*Maia et al., 2013a*). Distance between patches is proportional to the just noticeable differences (JNDs), assuming a UV-sensitive visual system (*Parrish et al., 1984*). (**B**) Proportional color variance among patches in an individual (violet), between sexes in a species (orange), and among species in a clade (green). Low variation between sexes was further confirmed with a multivariate phylogenetic integration test (r-PLS = 0.88, p < 0.01). Clades with more complex plumages (e.g., Alcedininae) tend to have a higher proportion of among-patch variation. (**C**) Distribution of multivariate evolutionary rates and (**D**) phylogenetic signal of color evolution across the body (darker colors indicate higher values).

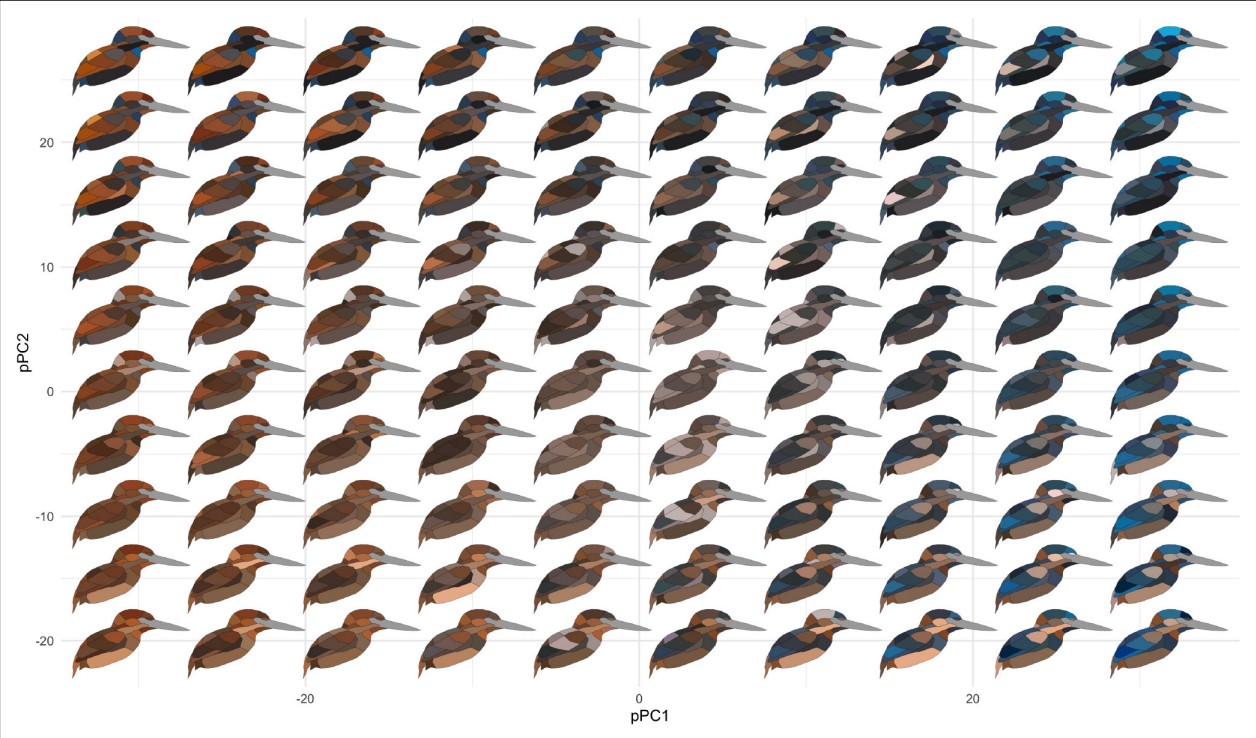

**Figure 3.** Color pattern morphospace of kingfishers. Bird images show depictions of color in a human visual system based on spectral measurements over a grid of phylogenetic principal components analysis (pPCA) coordinates. Axes shown are pPC axes 1 and 2, together accounting for >50% of plumage color variation in the clade.

The online version of this article includes the following figure supplement(s) for figure 3:

**Figure supplement 1.** Color pattern morphospace of kingfishers.

clades with complex color patterns that partition color variance more among patches than among species or individuals (e.g., *Corythornis,* Alcedininae) and (2) clades that vary more among species (e.g., *Todiramphus*, Cerylininae; *Figure 2B*). Chromatic variation among sexes was negligible for most clades (*Figure 2B*). Evolutionary rates of color were unevenly distributed across the body, with dorsal regions evolving faster than ventral ones (*Figure 2C*). This differs from several previous studies illustrating rapid rates of ventral plumage evolution in tanagers (*Shultz and Burns, 2017*), manakins (*Doucet et al., 2007*), fairy-wrens (*Friedman and Remeš, 2015*), and antbirds (*Marcondes and Brumfield, 2019*). This could indicate that dorsal plumage patches are under stronger sexual selection in kingfishers, as rapid rates of display trait evolution are thought to be associated with more intense sexual selection (*Irwin et al., 2008*; *Seddon et al., 2013*; *Merwin et al., 2020*). Rump, cheek, and throat patches showed the highest levels of phylogenetic signal (*Figure 2D*), suggesting that these patches are more taxonomically informative than crown or wing plumage coloration. To visualize major axes of variation in overall plumage color patterning, we used a phylogenetic principal components analysis (pPCA), with per-patch color coordinates as variables ($N$ = 66). We plotted the first two pPC scores that together accounted for >50% of color variation in the clade, revealing extensive color pattern variation in the group (*Figure 3*; see *Figure 3—figure supplement 1* for non-phylogenetic PCA results).

## A novel approach for estimating plumage complexity

To test our hypothesis that intraspecific plumage complexity facilitates interspecific color divergence, we required species-specific estimates of plumage color complexity. For each species, we calculated plumage complexity for both chromatic (i.e., hue and saturation) and achromatic components (i.e., lightness) of plumage patches in three ways: (1) as the mean pairwise distance among all patches in colorspace; (2) as the color volume (or lightness range for achromatic plumage components; see Methods) enclosing all points for a species, and (3) as the number of uniquely colored contiguous

patches on the body, assessed using just noticeable differences (JNDs >1 threshold) for a folded-wing plumage configuration (see *Figure 1A*, section 3b). The latter two metrics are similar to a recent method (*Eliason et al., 2019*) of calculating color complexity of plumages as the number of contiguous body regions sharing the same color mechanism (e.g., melanin-based or structural coloration), but they are based on continuous reflectance values instead of discrete color data (i.e., presence or absence of a given color mechanism). With this metric, higher differences between adjacent patches yielded higher plumage complexity scores (see *Figure 1A*, section 3b). Estimates of plumage complexity were strongly correlated among different complexity metrics for chromatic components of plumage coloration, but less so for achromatic variation (*Figure 1—figure supplement 1*).

## Species with complex plumages have higher rates of color evolution

Plumage complexity of an individual bird and interspecific differences in coloration are typically thought of as distinct axes of color diversity. Yet, species that have evolved several patches have more degrees of freedom to vary, potentially leading to faster rates of color evolution among species. However, this is not necessarily the case, as there are examples within kingfishers that show simple plumages yet high color divergence, as well as complex plumages with little evolutionary divergence (*Figure 1B*). Here, we attempt to link plumage complexity with interspecific rates of color variation using multivariate approaches typically only applied in the field of geometric morphometrics. To determine rates of overall plumage evolution, we used a recent time-calibrated phylogeny (*McCullough et al., 2019*) that incorporated thousands of ultraconserved elements (*Faircloth et al., 2012*) and fully sampled the avian order Coraciiformes (kingfishers, bee-eaters, rollers, and allies). Next, using multivariate color data, we estimated species-specific multivariate rates of evolution using the R package RRphylo v.

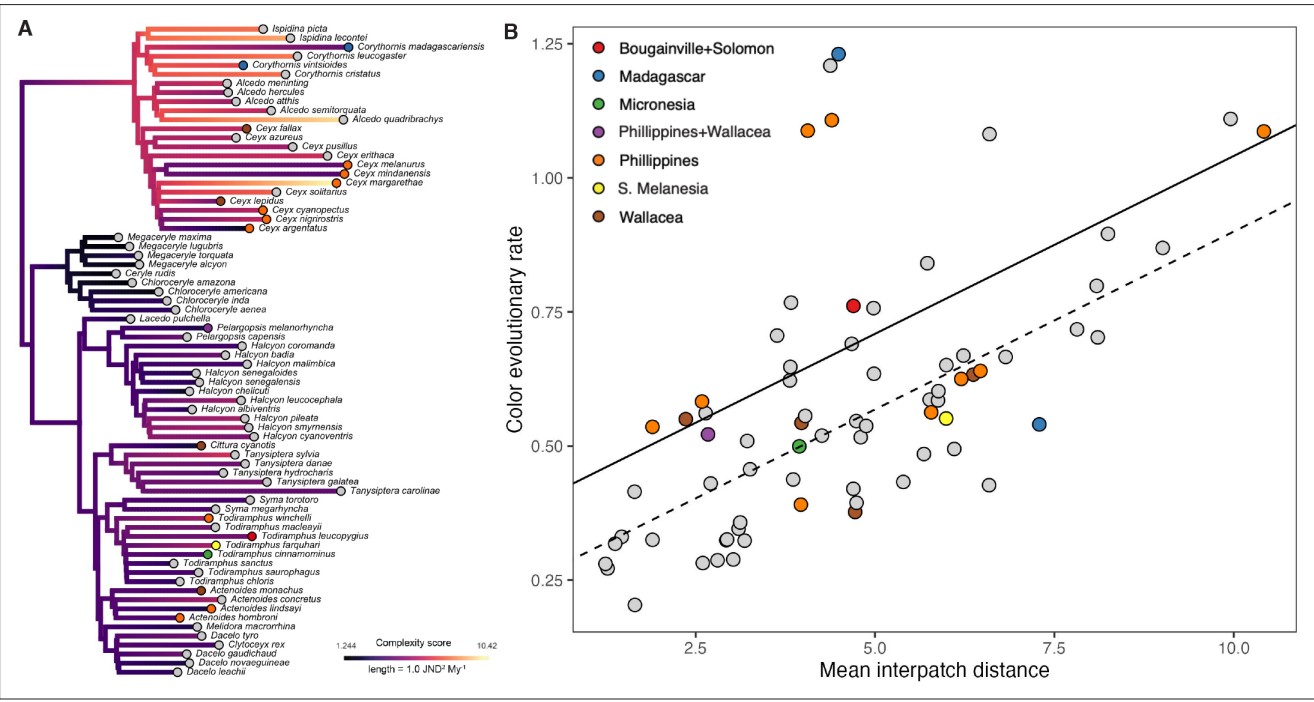

**Figure 4.** Species with complex plumages have faster rates of color evolution. (**A**) Phylogeny showing evolution of plumage color complexity, with edge colors corresponding to ancestral states of plumage complexity (mean interpatch color distance within a species, corresponding to metric c1 in *Figure 1*) and edge lengths proportional to color evolutionary rates (see legend). Tip colors correspond to different island systems (see legend in B), with continental species in gray. (**B**) Significant relationship between color evolutionary rates and plumage complexity (p < 0.01). Effect of island-dwelling (p = 0.02) is indicated by line type (dashed: continental, solid: island species). See *Table 1* for statistical results and *Figure 4—figure supplement 3* for results with analyzing achromatic variation in plumage.

The online version of this article includes the following figure supplement(s) for figure 4:

**Figure supplement 1.** Full PGLS results compared to various submodels explaining chromatic and achromatic rate variation.

**Figure supplement 2.** Islands and plumage complexity.

**Figure supplement 3.** Achromatic rate variation is associated with lightness range (metric c2) but not insularity.

**Table 1.** Plumage complexity predicts rates of color evolution among species.
Models were fit using PGLS in the phylolm R package, with species-specific evolutionary rates as the response variable and complexity metrics (c1, c2, and c3), island-dwelling, natural log body mass, and number of sympatric species as predictors. The best-fitting models were determined using a stepwise AIC-based procedure using the phylostep function in phylolm. Significant predictors in the most parsimonious models are indicated in bold. See *Supplementary file 1b* for sex-specific results and *Figure 4—figure supplement 1* for results with the full model and alternate submodels.

| Response | Predictor | Effect | p | $\lambda$ | $R^2$ |
|---|---|---|---|---|---|
| Chromatic rate | **Mean interpatch distance (c1)** | **0.41 ± 0.13** | **<0.01** | 0.00 | 0.45 |
| | # unique patches (c3) | 0.24 ± 0.13 | 0.07 | ... | ... |
| | **Insularity** | **0.49 ± 0.20** | **0.02** | ... | ... |
| Achromatic rate | **Lightness range (c2)** | **0.31 ± 0.11** | **<0.01** | 0.00 | 0.14 |
| | **ln body mass** | **0.23 ± 0.11** | **0.05** | ... | ... |

2.6.3 (*Castiglione et al., 2018*). Because we predicted that insularity results in faster rates of plumage color evolution, we included insularity as a covariate in our phylogenetic analyses. Comparing species-specific rates of plumage color evolution with intraspecific complexity metrics, we found that rates of color evolution were higher in species with more complex plumages (*Figure 4B*, *Table 1*; see *Supplementary file 1b* for sex-specific results). For achromatic variation, body mass and lightness range (c2) significantly explained increases in rates, but folded-wing achromatic complexity (c3) did not (*Table 1*). Although complexity metrics were correlated (*Figure 1—figure supplement 1*), variance inflation factors (VIFs) were not extreme (all <5), and phylogenetic generalized least squares (PGLS) results were stable after dropping each complexity variable from the reduced models (*Figure 4—figure supplement 1*). These results were further confirmed using a well-established multivariate method for comparing lineage-specific rates (*Denton and Adams, 2015*) based on binary complexity scores (*Supplementary file 1c*; see Methods for details).

Taken together, our findings are consistent with the idea of multifarious selection providing more axes for ecological or phenotypic divergence in complex color signals among species, and can eventually lead to speciation (*Nosil et al., 2009*). However, recent work in wolf spiders has revealed that signal complexity per se can be a direct target of sexual selection (*Choi et al., 2022*). Another possibility in kingfishers is that body size is driving the evolution of plumage complexity, as signal complexity has been shown to decrease with body size in iguanian lizards (*Ord and Blumstein, 2002*) and in passerine birds (*Cooney et al., 2022*). Interestingly, the kingfisher species with the most complex plumages are also among the smallest birds in the family, the pygmy-kingfishers, such as the indigo-banded kingfisher (*Ceyx cyanopectus*) and southern silvery-kingfisher (*C. argentatus*, *Figure 4A*). We

**Table 2.** Predictors of plumage complexity.
Models were fit for both chromatic (i.e., hue and saturation) and achromatic variables (i.e., plumage lightness) using PGLS in the phylolm R package. Different complexity metrics (see *Figure 1* for details) were set as the response variable, and island-dwelling, ln body mass, and number of sympatric species were used as predictors. The best-fitting models were determined using a stepwise AIC-based procedure (i.e., using the phylostep function). Significant predictors are indicated in bold. See *Supplementary file 1d* for sex-specific results.

| Data type | Response | Predictor | Effect | p | $\lambda$ | $R^2$ |
|---|---|---|---|---|---|---|
| Chromatic | Interpatch dist. (c1) | **ln mass** | **−0.34 ± 0.14** | **0.02** | 0.41 | 0.08 |
| | Color volume (c2) | **ln mass** | **−0.33 ± 0.14** | **0.02** | 0.18 | 0.07 |
| | # unique patches (c3) | **ln mass** | **−0.31 ± 0.15** | **0.05** | 0.51 | 0.05 |
| Achromatic | Lightness range (c2) | **Insularity** | **−0.60 ± 0.27** | **0.03** | 0.00 | 0.11 |
| | Lightness range (c2) | **# symp. species** | **−0.32 ± 0.12** | **0.01** | ... | ... |
| | # unique patches (c3) | Insularity | −0.41 ± 0.26 | 0.12 | 0.15 | 0.03 |

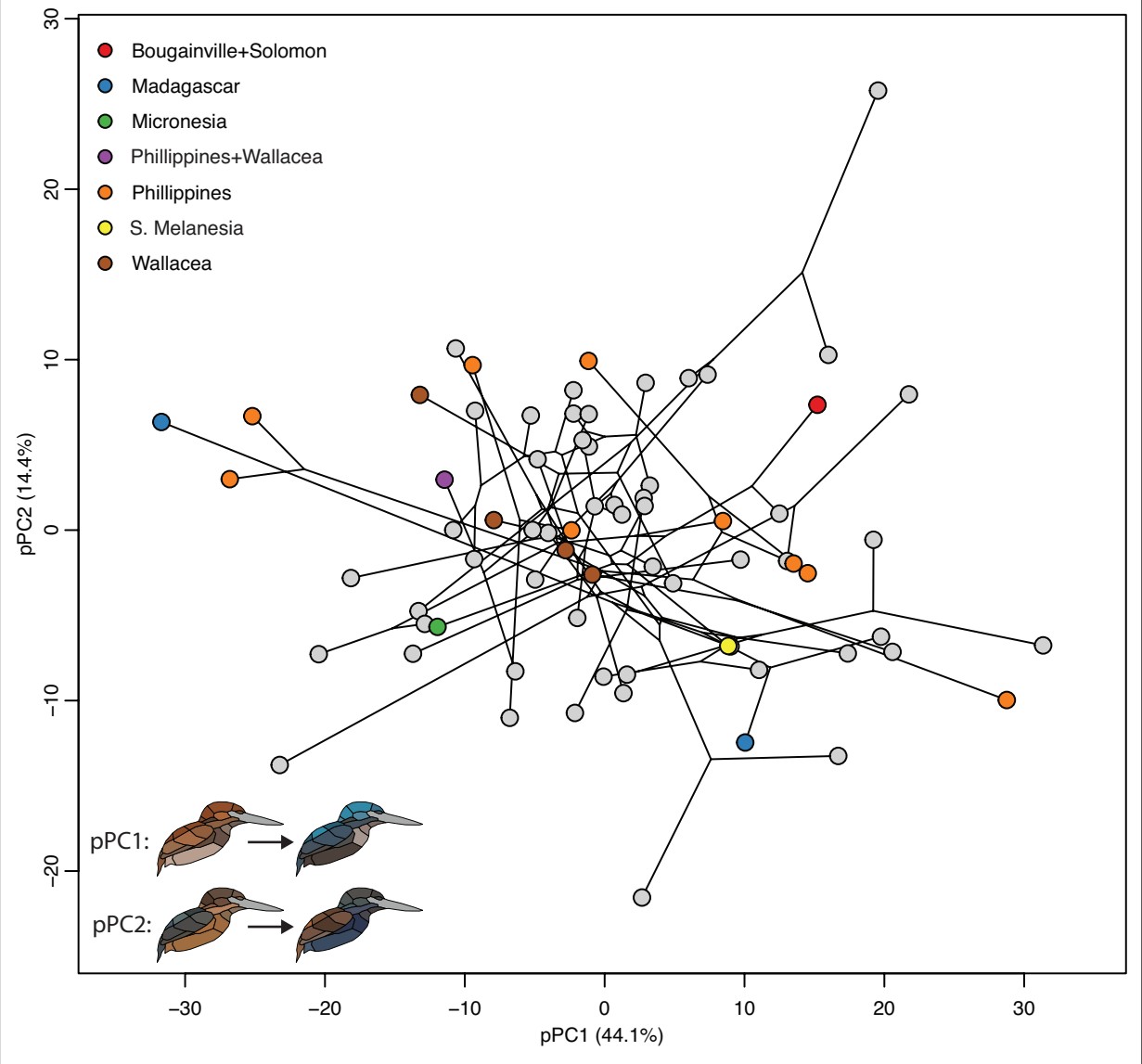

**Figure 5.** No support for convergence of color patterns on islands. Phylogenetic principal components analysis (pPCA) plot with points colored by continental (gray) and island species (see legend). Distance-based PGLS analyses suggest island and mainland species are not significantly different in plumage coloration (*F* = 1.84, p = 0.09). See inset for interpretation of pPC values and *Table 3* for full statistical results.

found some support for this hypothesis, as most chromatic complexity metrics were significantly lower in large-bodied species, whereas achromatic complexity was not linked to body size (*Table 2*). An alternative hypothesis is that species on islands have more complex plumages, and therefore insularity is indirectly driving color divergence. However, plumage complexity metrics were not significantly divergent between islands and mainland taxa (*Figure 4—figure supplement 2*, *Table 2*), suggesting that insularity and plumage complexity are independent drivers of color variation in the group.

## Island kingfishers have higher rates of color evolution

Colonization of islands is expected to result in shifts in both the direction of phenotypic change (i.e., convergent evolution when species colonize islands) and also the magnitude of change (i.e., elevated rates of phenotypic evolution on islands versus the mainland; *Millien, 2006*). To test these ideas, we first evaluated whether islands act as distinct selective regimes that drive convergent change toward particular colors using a distance-based PGLS (d-PGLS) approach developed for morphometric data (*Adams, 2014a*), but suitable for color data as well. Results of this analysis showed weak support for

**Table 3.** Multivariate plumage color is not significantly different on islands.
Results of multivariate distance-based PGLS (d-PGLS) tests testing for convergence in overall plumage coloration on islands. Both chromatic (i.e., hue and saturation) and achromatic plumage variables (i.e., lightness) were considered. p values were calculated with a permutation approach using 999 iterations. See *Figure 5* for details and *Supplementary file 1e* for sex-specific results.

| Response | Predictor | *F* | p | $N_{traits}$ | $N_{species}$ |
|---|---|---|---|---|---|
| Multivariate color | Insularity | 1.84 | 0.09 | 66 | 72 |
| | # sympatric species | 0.35 | 0.94 | ... | ... |
| | ln body mass | 1.14 | 0.32 | ... | ... |
| Multivariate lightness | Insularity | 0.83 | 0.59 | 22 | 72 |
| | # sympatric species | 1.14 | 0.32 | ... | ... |
| | ln body mass | 1.67 | 0.11 | ... | ... |

the prediction that island colonization has caused convergent evolution of color (p = 0.09; *Figure 5*, *Table 3*), while lightness showed no significant difference between mainland and island species (p = 0.59; *Table 3*). This is distinct from a previous study showing predictable trends toward darker plumages on islands (*Doutrelant et al., 2016*). However, we did find that achromatic complexity was significantly lower on islands (*Table 2*).

To test for an 'island effect' on rates of color evolution, we treated individual patches as geometric morphometric 'landmarks' and compared multivariate evolutionary rates between insular and continental species using rate ratio tests (*Denton and Adams, 2015*). When considering the island effect alone on rates of color evolution, we found that species distributed on islands have faster rates of color evolution ($\sigma^2_{cont}$ = 0.13, $\sigma^2_{island}$ = 0.23, p = 0.02; *Figure 4A*) but similar rates of light-to-dark evolution compared to continental species ($\sigma^2_{cont}$ = 0.84, $\sigma^2_{island}$ = 0.91, p = 0.72). To further test the possibility that the observed rapid color evolution on islands is the result of reproductive character displacement occurring within islands (e.g., see *Losos and Ricklefs, 2009*), we included the number of sympatric kingfisher species as a predictor in our PGLS models. The number of sympatric lineages ranged from 1 to 9 on islands, and 6–38 for mainland taxa (see Dryad). Neither overall plumage color patterns (*Table 3*) nor rates of plumage evolution (*Table 1*) were significantly associated with the number of sympatric species. Thus, rather than interspecific competition driving color diversity, intraspecific competition or genetic drift may instead be driving rapid rates of color evolution in island kingfishers.

## Discussion

We lack a cohesive understanding of how plumage color patterns evolve in birds. This study is the first attempt to link intraspecific color variation among patches to interspecific color variation among species. We find support for higher rates of plumage evolution in clades with more complex plumages (*Figure 4*). This supports the idea that plumage complexity, rather than uniformity, provides more phenotypic traits for natural selection to act upon. In addition, we find that island lineages have faster rates of plumage evolution (*Table 1*), but not more complex plumages (*Table 2*, *Figure 4—figure supplement 2*), than continental lineages.

Colonization of novel geographic areas can promote either shifts in mean phenotype or changes in rates of phenotypic evolution (*Collar et al., 2009*). Changes in rates associated with island colonization have been described in lizards (*Pinto et al., 2008*) and mammals (*Millien, 2006*). *Pinto et al., 2008* found that rates of morphological evolution were not elevated in Caribbean anoles compared to mainland species, but they did show differences in morphospace (i.e., convergence). In birds, *Doutrelant et al., 2016* measured coloration in 4448 patches of 232 species (including eight kingfisher species) and found that island-dwelling species have darker colors and fewer color patches (i.e., less complex plumages) than mainland species. This differs from the results of our study, as we found no difference in achromatic (*Table 3*) or chromatic plumage complexity between mainland and island species (*Table 2*, *Figure 4—figure supplement 2*). Rather, it is the rates of evolution that

increase once kingfishers colonize islands (*Figure 4B*). This suggests decoupling between the effects of complexity and insularity on color evolution rates, and is consistent with previous work showing elevated rates of morphological evolution being independent of the acquisition of a key innovation, such as gecko toepads (*Garcia-Porta and Ord, 2013*). However, this does not answer the question of why plumage color among island lineages would differ more than among mainland lineages.

Contrary to the prevailing view that island species should have reduced diversity of mate recognition signals (*West-Eberhard, 1983*), island kingfishers have more variable plumages than their continental relatives (*Figure 4B*, *Table 1*). Other examples of this pattern of elevated signal diversity on islands include *Anolis* lizards (*Gorman, 1968*) and *Tropidurus* lizards (*Carpenter, 1966*) that both show high diversity in dewlap displays. Historical explanations for why colors might evolve include increased conspicuousness for mating displays and more efficient species recognition (*Andersson, 1994*; *Doucet et al., 2007*). The species-recognition hypothesis predicts reduced signal distinctiveness (i.e., low amounts of plumage variation) on islands (*West-Eberhard, 1983*). This is because of the lack of potential competitors and conspecifics on islands that would otherwise put selective pressures on color patterns, for example through reproductive character displacement (*Drury et al., 2018*). However, we found no support for this idea, as evolutionary rates of coloration were not significantly associated with the number of sympatric species (*Table 1*). Another mechanism that could explain the observed rapid color evolution on islands is divergence in abiotic factors among islands. Structural coloration, responsible for vivid blues, greens, and purples, is salient in kingfishers (*Stavenga et al., 2011*; *Eliason et al., 2019*). Compared to pigment-based colors, structural colors exhibit some of the fastest rates of color evolution known in birds (e.g., hummingbirds; *Eliason et al., 2019*; *Venable et al., 2022*). Structural and melanin-based forms of coloration may have thermal benefits to birds (*Rogalla et al., 2022*), and both molt speed (*Griggio et al., 2009*) and dietary protein availability (*Meadows et al., 2012*) have been shown to influence structurally colored signals. Thus, divergence in food availability or climate among island populations could be driving rapid shifts in coloration. Future work will be needed to tease apart the relative roles of genetic drift, competition, and abiotic factors in driving color evolution on islands. The kingfisher genus *Todiramphus* harbors several 'superspecies'—monophyletic groups of allopatric and morphologically distinct taxa (*Mayr, 1963*)—and therefore could be an ideal study system for clarifying the roles of ecology and constraint in driving color diversity on within and between islands.

Studying color pattern evolution has been historically difficult, due, in part, to an inability of humans to perceive UV color (*Eaton, 2005*) and difficulties with measuring and analyzing complex color patterns (*Mason and Bowie, 2020*). Recent work has showed that changes in plumage complexity are associated with shifts in light environment (*Maia et al., 2016*; *Shultz and Burns, 2013*). Here, plumage complexity was treated as a response variable rather than as a predictor of overall color divergence between lineages. Our results therefore provide a contrast to previous work in looking at a potential developmental constraint—how plumage patches are arranged on the body (*Price and Pavelka, 1996*)—and its causal influence on evolutionary trends of color divergence. A caveat with our approach is that it does not consider color patterning within feathers. For example, the species with the least complex plumage according to the mean interpatch color distance (metric $c_1$) is the pied kingfisher (*Ceryle rudis*), despite its conspicuous black and white barring/spotting across its body and even within individual feathers. We hope that researchers will consider the morphometrics approach we take here, as well as assess its potential strengths and weaknesses, in future studies on the evolution of complex color patterns in nature.

In this study, we collected a large amount of spectral data (9362 measurements of 142 individuals in 72 species) in a diverse family of birds notable for their complex plumages and rapid speciation on islands. The major finding of our study is that complex plumage patterns enable faster rates of color evolution compared to simpler, uniform plumages. Colonization of islands, independent of plumage complexity, resulted in further divergence of coloration among species. More broadly, these results highlight the interplay between a potential key innovation (i.e., plumage complexity) and geographical opportunity for allopatric speciation in birds. It also highlights the need for incorporating multidimensional aspects of plumage patterns in such analyses. Further research is needed to test whether complex plumages are more common in clades that are speciating rapidly and if complexity is itself a direct target of sexual selection (e.g., *Choi et al., 2022*).

# Materials and methods

## Measuring feather color

Using a UV–Vis spectrophotometer (Ocean Optics) operating in bird-visible wavelengths (300–700 nm), we measured reflectance spectra at normal incidence in triplicate for 22 patches (see *Figure 1—figure supplement 2*) in 72 species, including both males and females, from museum specimens. In total, we obtained 9303 spectra for 142 individuals (available on Dryad). We averaged three spectra per patch per specimen and converted values into avian tetrahedral colorspace (u, s, m, and l channels) using the vismodel function in pavo (*Maia et al., 2013a*), based on a UV-sensitive visual system (*Parrish et al., 1984*). We converted quantum catches into perceptually uniform *XYZ* coordinates (*Pike, 2012*) for use in downstream comparative analyses (*Figure 2A*). The distance between pairs of coordinates in this colorspace is proportional to the just noticeable difference (JND). As these data only capture variation in chromatic aspects of coloration, we also assessed achromatic variation by calculating luminance as the quantum catch for the double cone. We used photosensitivity data for the blue tit (*Hart et al., 2000*) due to the limited availability of sensitivity data for other avian species, and we further accounted for receptor noise following *Olsson et al., 2018*. We converted luminance values into a scale where distances between pairs of measurements are equivalent to JND values by subtracting ln(0.01) from the ln luminance values and dividing this by the Weber fraction ($\omega = 0.1$), following *Pike, 2012*. Although it is possible, in theory, to combine chromatic and achromatic channels of plumage variation in a single analysis (*Pike, 2012*), we opted to analyze them separately because they are likely under different selection pressures (*Osorio and Vorobyev, 2005*) and we wanted to be able to compare our results with previous work on island bird coloration (*Doutrelant et al., 2016*).

## Assessing color variation

To gain an understanding of how color varies across different levels of organization (e.g., plumage patches), we performed a taxonomic analysis of variance (*Derrickson and Ricklefs, 1988*). Briefly, we fit a linear mixed model in MCMCglmm (*Hadfield and Nakagawa, 2010*) using colorspace *XYZ* coordinates as amultivariate response, with random effects for plumage patch, sex, and species. We set rather uninformative priors ($v = 2$ for random effects, $v = 0.002$ for residual covariance) and ran the Markov chain Monte Carlo (MCMC) chains for $10^6$ generations, discarding 25% as burn-in. We ran two chains and assessed convergence by plotting the Gelman–Rubin diagnostic (*Gelman and Rubin, 1992*) using gelman.plot in the R package coda 0.19.4. From the fitted models, we calculated the sum of mean posterior variances for each colorspace coordinate and estimated the proportional amount of variation explained at each level of organization by dividing each variance by the total variance (see Dryad for R code). To visualize color pattern diversity, we used pPCA to reduce the dimensionality of our color data set. We performed pPCA using the phyl.pca_pl function in R (*Clavel et al., 2019*) based on the covariance matrix. Phylogenetic PCA has been criticized because of the influence of component selection bias when used in downstream comparative analyses (*Uyeda et al., 2015*), therefore we also performed an ordinary PCA with the prcomp function in R, with similar parameters as above.

## Quantifying plumage complexity

To estimate plumage complexity at the individual level, we obtained pairwise perceptual distances (proportional to just noticeable differences, JNDs) using the coldist function in pavo (*Maia et al., 2013b*). Since we measured color of 22 patches, this resulted in a 22 × 22 matrix for each individual. Next, using these chromatic and achromatic distances, we calculated plumage complexity in three ways (see *Figure 1A*, section 3b). First, we averaged these distance matrices by species and calculated mean interpatch distances (metric c1) using the R dist function. Second, we calculated the total colorspace volume occupied by an individual's plumage (metric c3) as a 3D volume for *XYZ* colorspace coordinates. We used the convhulln function in the R package geometry v. 0.4.6.1 to calculate this metric, which can also be described as the range of luminance values for achromatic variables. Third, we calculated complexity as the number of distinct contiguous patches in folded-wing body configuration (metric c3). To do so, we converted species color distance matrices to binary scores indicating whether pairs of patches are perceptibly distinct (JND >1) or not (JND <1), resulting in a pairwise color distance matrix ($M_{JND}$), with 0 indicating patches that are distinct and 1 indicating patches that would be perceived as the same by a bird. For each individual, we multiplied $M_{JND}$ by the adjacency matrix $M_{adj}$. In $M_{adj}$, 1 indicates patches that are adjacent on a bird's body and 0 indicates

non-adjacent patches. Multiplying $M_{JND}$ by $M_{adj}$ results in a matrix with 1 if patches are both adjacent and indistinguishable in colorspace (*Figure 1A*). Finally, we converted these final matrices into igraph objects using graph_from_adjacency_matrix and determined the number of distinct plumage regions using the components function in igraph (*Csardi and Nepusz, 2006*). We only considered a folded-wing plumage configuration because this is how a bird would be typically seen by a conspecific and because folded-wing complexity scores were highly correlated with spread-wing complexity ($r = 0.98$).

## Understanding the tempo and mode of color evolution

To understand evolutionary trends on a per-patch basis, we compared phylogenetic signal and rates of color evolution within each individual plumage patch ($N = 22$, *Figure 2C, D*) using distance-based comparative methods (*Adams, 2014b*; *Denton and Adams, 2015*). Next, to account for phylogenetic signal at the overall plumage level, we fit multivariate Brownian motion and Pagel's $\lambda$ models for all color traits (i.e., 22 lightness variables, 66 color variables) using the fit_t_pl function (*Clavel et al., 2019*) in RPANDA v. 2.1 (*Morlon et al., 2016*). We compared models using generalized information criteria. The best-fitting model was a Pagel's $\lambda$ model for both achromatic and chromatic plumage components (*Supplementary file 1a*), thus we used these $\lambda$ estimates to transform branch lengths of the phylogeny prior to running multivariate comparative analyses. Although variance tests revealed some color variation attributable to sex differences (*Figure 2B*), multivariate phylogenetic integration tests (*Adams et al., 2014c*) revealed significant correlations between male and female plumage coloration for both plumage color (r-PLS = 0.88, $p < 0.01$, $N = 57$) and lightness (r-PLS = 0.93, $p < 0.01$, $N = 57$). Therefore, for primary analyses, we used a combined data set with male and female color data averaged together for each species. However, we also included results for the males-only ($N = 63$) and females-only data sets ($N = 68$; see *Supplementary file 1a–e*).

## Testing ecological predictors of plumage color variation

To the prediction that species on islands have less complex plumages, as predicted by the species-recognition hypothesis (*West-Eberhard, 1983*), we used phylogenetic linear models (*Ho and Ané, 2014*). For the models, complexity metric was set as the response variable and the predictors were insularity and the number of sympatric species (additional parameters: method = "lambda", lower. bound = 1e−10; see Dryad for R code). Body mass has recently been shown to explain variation plumage complexity of passerine birds (*Cooney et al., 2022*), therefore we also included ln body mass (in grams) as a covariate in our regression models (species averages obtained from *Dunning, 2007*). We removed non-important variables from the models using a bidirectional AIC-based step-wise procedure in phylostep. To further test whether species on islands are convergent in their overall plumage color patterns, we used a multivariate d-PGLS approach (*Adams, 2014b*) implemented in the prodD.pgls function of geomorph v. 4.0.4 (*Adams et al., 2013*). For d-PGLS models, either multivariate color or multivariate lightness was set as the response variable, and the predictor variables were insularity, ln body mass, and the number of sympatric species. We assessed significance using a permutation approach with 999 iterations.

## Determining drivers of evolutionary rate variation in plumage color

To test our predictions that rapid rates of color evolution are associated with complex plumages and insularity, we again used PGLS models implemented in the phylolm R function (*Ho and Ané, 2014*). We ran PGLS models with species-specific rates as the response and complexity metrics ($c_1$, $c_2$, and $c_3$), ln body mass, insularity, and the number of sympatric species as predictors. We included several complexity metrics in the same analyses since these metrics are likely capturing somewhat different aspects of plumage complexity. As an example, a hypothetical bird with a mostly black plumage and a single red patch would result in a high color volume despite a low mean interpatch distance. We fit models for both plumage variable types (chromatic and achromatic) and sexes (males and females) using the options method = "lambda" and lower.bound = 1e−10 and determined the best-fitting model using a stepwise AIC-based procedure with phylostep (see *Supplementary file 1f and g* for results of all models tested). Because some complexity metrics were correlated (*Figure 1—figure supplement 1*), we assessed multicollinearity with VIFs. Briefly, following *Mundry, 2014*, we (1) re-formulated the best-fitting PGLS model with each complexity metric set as the response variable rather than a predictor, (2) estimated $R^2$ values for each model with the R2.lik function in rr2 v. 1.0.2 (*Ives,*

*2019*), and (3) calculated VIFs as $1/(1R^2)$. To further test the sensitivity of PGLS estimates to potential multicollinearity, we re-fit the best-fitting models after dropping each complexity metric in turn (see *Figure 4—figure supplement 1* for results). All variables were scaled prior to PGLS analysis to make coefficients comparable across models (i.e., as effect sizes).

In addition to this PGLS approach, we also used a well-established method (*Denton and Adams, 2015*) for comparing rates among groups with high and low plumage complexity scores. This analysis required binary estimates of complexity, therefore we used kmeans clustering (centers = 2) to derive binary complexity scores. We then input these values as a predictor of rate variation into the compare. evol.rates function (*Denton and Adams, 2015*), with multivariate color ($N$ = 66 traits) or lightness ($N$ = 22) as the response. We used the permutation option with 999 iterations to assess significance of the relationship between predictors (complexity or insularity) and rates of color evolution. We used a similar approach with insularity as a predictor of rate variation.

## Acknowledgements

We thank Ben Marks for assistance with bird specimens at the FMNH. We also thank Kristopher Menghi who helped with the collection of spectral data. This work was partially supported by grants from the National Science Foundation (NSF EP 2112468 to CME and SJH, NSF EP 2112467 and DEB 1557051 to MJA).

## Additional information

### Funding

| Funder | Grant reference number | Author |
|---|---|---|
| National Science Foundation | EP-2112468 | Chad M Eliason |
| National Science Foundation | EP-2112467 | Michael J Andersen |
| National Science Foundation | DEB-1557051 | Michael J Andersen |

The funders had no role in study design, data collection, and interpretation, or the decision to submit the work for publication.

### Author contributions

Chad M Eliason, Conceptualization, Data curation, Formal analysis, Funding acquisition, Writing – original draft; Jenna M McCullough, Writing – original draft; Shannon J Hackett, Michael J Andersen, Funding acquisition, Writing – original draft

### Author ORCIDs

Chad M Eliason ⬤ http://orcid.org/0000-0002-8426-0373
Jenna M McCullough ⬤ http://orcid.org/0000-0002-7664-3868

### Decision letter and Author response

Decision letter https://doi.org/10.7554/eLife.83426.sa1
Author response https://doi.org/10.7554/eLife.83426.sa2

## Additional files

### Supplementary files

• Supplementary file 1. Supplementary tables. (a) Evolutionary model fits using multivariate color data sets. (b) Predictors of rates of color evolution when analyzing males and females separately. (c) Evolutionary rates tests using discrete plumage complexity scores. (d) Predictors of plumage complexity for males and females analyzed separately. (e) Testing predictors of shifts in average plumage hue and brightness. (f) Full PGLS regression results for predictors of chromatic rate

variation. (g) Full PGLS regression results for predictors of achromatic rate variation.

- MDAR checklist

### Data availability

Data sets and R code used in analyses within the manuscript are available at: https://doi.org/10.5061/dryad.5mkkwh78v.

The following dataset was generated:

| Author(s) | Year | Dataset title | Dataset URL | Database and Identifier |
|---|---|---|---|---|
| Eliason C, McCullough J, Hackett S, Andersen M | 2023 | Complex plumages spur rapid color diversification in kingfishers (Aves: Alcedinidae) | https://doi.org/ 10.5061/dryad. 5mkkwh78v | Dryad Digital Repository, 10.5061/dryad.5mkkwh78v |

The following previously published datasets were used:

| Author(s) | Year | Dataset title | Dataset URL | Database and Identifier |
|---|---|---|---|---|
| Eliason C, Andersen M, Hackett S | 2020 | Data from: Using historical biogeography models to study color pattern evolution | https://doi.org/10. 5061/dryad.3680n0c | Dryad Digital Repository, 10.5061/dryad.3680n0c |
| Eliason C, McCullough J, Andersen M, Hackett S | 2021 | Accelerated brain shape evolution is associated with rapid diversification in an avian radiation | https://doi.org/10. 5061/dryad.ffbg79cs6 | Dryad Digital Repository, 10.5061/dryad.ffbg79cs6 |

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
