## [Editor Report]

This important work advances our understanding of the factors that affect the speed of colour evolution in birds and the resulting diversification patterns. It provides compelling evidence that more complex plumage coloration can lead to rapid colour evolution in kingfishers, and will pave the way for more comprehensive analyses that fully embrace the multidimensional nature of colour variation. Hence, the results will be of broad interest to ornithologists and evolutionary biologists in general.

---

## [Decision Letter]

**Decision letter after peer review:**

Thank you for submitting your article "Complex plumages spur rapid color diversification in island kingfishers (Aves: Alcedinidae)" for consideration by *eLife*. Your article has been reviewed by 3 peer reviewers, including Kaspar Delhey as Reviewing Editor and Reviewer #1, and the evaluation has been overseen by Christian Rutz as the Senior Editor.

Essential revisions:

1) The two main topics linked to evolutionary rates of plumage colour, island-dwelling, and plumage complexity need to be better introduced and linked in the introduction. This is a point that was made in one form or another by the three reviewers.

2) The concept of plumage complexity, which is central to the paper, requires a more comprehensive explanation. In particular, how it differs from evolutionary divergence to avoid any potential circularity.

3) The methods used to analyse differences in colour evolutionary rates constitute some of the most novel aspects of the paper. However, these are difficult to follow. The authors need to cater for two different types of readers here:

(a) Those that only want to broadly understand the methods used and

(b) those that will want to adapt the methods to their own study system.

To deal with (a) Rev 4 suggestion of a flowchart figure illustrating the methods provides an elegant way to summarise the approach used. For (b) much more detail is needed regarding the multiple ways in which analyses can vary, this ranges from information regarding arguments used within specific functions to better explanations as to why certain methodological decisions were taken.

4) There are many statistical analyses in the paper. Some may be better suited than others (e.g. the pitfalls of using PCA suggest that maybe they are less well suited) and some seem somewhat redundant. The authors should try to streamline this as much as possible, potentially discarding ancillary analyses that do not deal with the main questions.

*Reviewer #1 (Recommendations for the authors):*

Specific comments (ln refer to line numbers)

Ln 112-128 Is the PCA needed to describe colour variation? It does not seem to add much and PCA has been recently criticised in the context of comparative analyses and other multivariate approaches (some of which are implemented in the paper) are preferred (Uyeda et al., 2015; Adams and Collyer, 2018).

Ln 139 high phylogenetic signal == species diagnostic traits. I am not sure that this should be the case, the high phylogenetic signal would indicate that closely related species have similar colours, and hence such species groups would share rather than differ in these traits. Thus, species identity among closely related species would not be easily signalled by such colours.

Ln 179-180 and in passerine birds see (Cooney et al., 2022).

Ln 140-187 Why did you not try to use geomorph::compare.evol.rates to compare multivariate evolutionary rates between species with low and high plumage complexity? I know this approach only deals with tips and not nodes, but it is a bit unsatisfactory to use one approach for one question and another one for the other given that the procedures are quite different.

Ln 225 why intra-specific competition?

Ln 236 p=0.08 should not be interpreted as no support (see (Muff et al., 2021)), and it would be good to state the direction of this effect in the text as well (does it lean towards convergence or away from it?) as most readers will not be familiar with this particular statistic.

Ln 238 with 'drabber' do you mean darker (less light)? I would be careful with using terms such as drab or bright (dark and light would be better) without a clear definition first.

Ln 239 I am not sure that these results add much, the comparison between structural and pigmentary colours is a bit redundant (mechanisms were largely identified due to properties of the reflectance spectrum). I think that including these results detracts from the main message, similarly, I am not sure about the relevance of sex differences. Differences in mean colour between island and mainland species -if included- should come earlier in the more descriptive part of the results. Why did you not use geomorph::procD.pgls for this analysis? As an aside, how did you deal with sex differences in the other analyses? I might have missed it but, do they represent both males and females or only one sex? Please clarify.

Ln 280 "… competition between individuals of the same species is driving color diversity…" this needs further explanation, what would the mechanism be?

Ln 307-310 Why is your node+tips approach inherently better than a tips-only approach, other than by being more conservative? It could also be that including nodes increases type II error rates simply due to the inherent uncertainty associated with the estimation of ancestral traits (which often end up being an average of the tips). I think that a bit more justification is needed here.

Ln 312 nor achromatic variation.

Ln 332 you averaged three spectra per patch per specimen, right?

Ln 344-46 PCA using a covariance or correlation matrix? Covariance would be the appropriate choice here as all variables are measured in the same units. This is not clear from the R code provided (at least to me). For the non-phylogenetic PCA probably the covariance matrix was used as this seems the default option for function prcomp, but I am not sure what the case is for the phylogenetic PCA function used. Nevertheless, this should be clear in the text.

Ln 346 pPCA has been criticised before see (Uyeda et al., 2015; Adams and Collyer, 2018), and this is particularly relevant when choosing to discard some PCs from subsequent analyses.

More information on the MCMCglmm priors is needed, from the code I get the impression that default priors were used (i.e. I see no prior argument specified in the code, or am I misunderstanding something)? Such default priors are usually not good for random effects. Also, it would be good to state that models converged and how this was assessed. Also, the multivariate Wald tests used in these MCMCglmm models should be better explained in the text.

The function geomorph::compare.evol.rates has two methods to establish whether evolutionary rate ratios between two regimes differ significantly from 1, the default ("simulation") and an alternative ("permutation"). In my experience using this function to analyse evolutionary rates of colour, "simulation" yields much lower p-values than "permutation" (unpubl. data). Based on reading (Adams and Collyer, 2018) and on direct communication with the main author of the geomorph package (Dean Adams) it seems that "permutation" is a better alternative to achieve an appropriate level of type I errors. Based on the R code provided it seems that the authors have used the default option "simulation". I think that the analyses using this function should be re-run with the alternative "permutation". I suspect that the evolutionary rate ratio will not differ significantly from 1 in that case. This should not matter, the result in itself is interesting regardless of the outcome, although it could mean that some key conclusions of the paper will have to be changed.

Figure 1. This figure could be deleted or moved to the suppl material, it does not add much to the general story.

Figure 3C It would be good to label data points that correspond to nodes vs tips in the figure to illustrate differences in results. Would it be possible to also include island/mainland dwellings in this analysis? It should be doable to reconstruct ancestral states for this trait as well right?

Figure 7 I am not sure this figure should be included in the main text, it is hard to visualize the patterns.

References

Adams, D.C. and Collyer, M.L. 2018. Multivariate Phylogenetic Comparative Methods: Evaluations, Comparisons, and Recommendations. Syst. Biol. 67: 14-31.

Cooney, C.R., He, Y., Varley, Z.K., Nouri, L.O., Moody, C.J.A., Jardine, M.D., et al. 2022. Latitudinal gradients in avian colourfulness. Nat. Ecol. Evol. 6: 622-629.

Muff, S., Nilsen, E.B., O'Hara, R.B. and Nater, C.R. 2021. Rewriting Results sections in the language of evidence. Trends Ecol. Evol. 37: 203-210.

Olsson, P., Lind, O. and Kelber, A. 2018. Chromatic and achromatic vision: parameter choice and limitations for reliable model predictions. Behav. Ecol. 29: 273-282.

Siddiqi, A., Cronin, T.W., Loew, E.R., Vorobyev, M. and Summers, K. 2004. Interspecific and intraspecific views of color signals in the strawberry poison frog Dendrobates pumilio. J. Exp. Biol. 207: 2471-2485.

Uyeda, J.C., Caetano, D.S. and Pennell, M.W. 2015. Comparative Analysis of Principal Components Can be Misleading. Syst. Biol. 64: 677-689.

*Reviewer #2 (Recommendations for the authors):*

Line 16: I think I know what you mean but this is not actually clear. By phenotypic, you mean increased rates of change per speciation event, correct? Whereas by species diversification you just mean increased speciation rates, correct? Make this clearer.

Line 25: I'm immediately wondering how plumage complexity is calculated and whether it de facto will tend to correlate with color diversity.

Lines 38-40: I'm not sure this is how the island rule is phrased. Isn't it more that organisms tend to shift notably in body size? Otherwise, how do we define a large-bodied colonist vs. a small-bodied colonist? Suggest making this sentence more general.

Line 58-59: I'm not sure what you mean here. If a color variant for a species with uniform coloration had higher fitness then yes, the whole plumage would change in tandem, but if the mutation involved a patch changing color, then the whole plumage wouldn't change in tandem. The way this is worded makes it sound like a uniformly colored species will be unable to evolve plumage complexity.

Line 63: are the authors going to define plumage complexity as the number of plumage patches that are different from one another? Because, if so, this will QED correlate with color variation. Revisiting this point after a few reads of the manuscript, I realize the authors are working across intraspecific and interspecific scales in this sentence. This needs revision here.

Lines 96-98: a lot of this feels really circular. Some attention is needed here. Cutting some words, this sentence says, "promote color divergence…contributing to high rates of color evolution and disparity". How color divergence differs from rates of color evolution differs from color disparity is not clear.

Line 122: change to "sex" in Figure 2, not "individual".

Line 127: I'm confused. Isn't Tanysiptera monomorphic?

Lines 124-128: this immediately makes me wonder whether type (i) tends to be allopatric more often than does type (iii). I think the answer is yes.

Line 137: This is in line with a recent Merwin et al. study on parrots (https://bmcecolevol.biomedcentral.com/articles/10.1186/s12862-020-1577-y).

Lines 138-139: High phylogenetic signal in X vs Y does not mean that X is "more diagnostic of species" than Y. Re-word to clarify.

Lines 152-157: Ok, it does seem like the authors define plumage complexity as the number of uniquely colored patches.

Lines 180-187: I think there are a few key results here that deserve more space.

Line 206: the result is not that they are de novo evolving new colors, correct? Rather, they are shifting around in color space more quickly. Clarify.

Lines 209-225: There are four substantial issues in this paragraph. First, anytime I hear about creating two separate trees from one to run a test I get leery. I'm not sure what the justification is here. Second, how many island species occur with a sympatric relative? I believe that the answer is not many, so we wouldn't expect interspecific competition to be relevant, correct? Third, I'm fairly sure these RPANDA functions can take a geography object to account for this varying sympatry/allopatry. No mention is made of whether this was done or, if so, how. Fourth, RPANDA isn't cited. I may be missing something, but I also believe the last sentence here is not warranted. It could also mean that drift on islands best explains the observed pattern.

Line 237: in what taxon?

Line 247: I'm confused. Earlier you said this author DID find predictable patterns towards duller plumage. You also say it again in the discussion below.

Line 289: What island is this? Does New Guinea count as an island?

Lines 273-293: I am not following the logic in this paragraph. I see no reason why not finding evidence for interspecific competition influencing plumage patterns makes any statement about the relevance of intraspecific competition. This is further confused by the first discussion in the manuscript of sympatry and allopatry, which is then not actually tied into the mechanisms of interest (competition and character displacement). This is a weak point of the paper as it stands.

Lines 300-309: I'm still a little confused about the differences between plumage complexity and evolutionary rates (it would be helpful to keep reminding the reader that one is within a species, and one is between species), but I think some of what the authors are saying here can be attributed to differences between a phylogenetic independent contrasts-esque approach (what the authors use), and a phylogenetic generalized least squares approach (what others tend to have used). The former tends to be more conservative in my experience. Per a comment above, I'm withholding too many thoughts about alternative tests until I get a clearer picture of what the authors want to test, but I suspect this could be done as a phylogenetic t-test, where the test is whether plumage complexity differs between island taxa or not.

Figure 7: very challenging to see. It might be clearer with a white background.

Figure S6: "Effect of data analysis on rate-complexity correlations". Data analysis is too vague. Can you come up with a more specific figure legend?

Line 43: errant parentheses.

Lines 92-94: run-on sentence, passive voice. Revise.

Line 103: missing a word after color.

Line 291: are->is

Line 411: the website for this paper incorrectly lists Ian PF Owens as just PF Owens. If you check the original manuscript, it correctly lists it as IPF Owens.

*Reviewer #3 (Recommendations for the authors):*

My line-by-line suggestions follow, I hope are helpful:

Line 15: This is entirely pedantic, but children typically leave the cradle, whereas oceanic island lineages often remain confined there until extinction or replacement (as in Wilson's taxon cycle). Maybe there's a better analogy out there.

Lines 57-58: Constraints on morphospace need not constrain evolutionary rates, see Goswami et al. 2014.

Line 60: Defining complexity can be a difficult topic to agree on in evolutionary biology and I think that considering the centrality of pattern complexity to the study, this would benefit from being discussed in more detail. Complexity at one level (phenotype) may not match with complexity at others (biochemistry, development, genome). I could see a working definition in the methods, as well as a supporting citation or two as a way to resolve this issue.

Lines 66-84: Can this paragraph be condensed and combined with the one below? I'm not seeing how it's relevant enough to the paper's key ideas to merit this much-lit review text. I don't want to hold this against the authors though, because I'm getting a subtle impression that a previous reviewer might have requested it, and nothing is more infuriating than having reviewers from different journals disagree with each other.

Line 82-84: I don't disagree with this outright – it does seem to be the case in starlings – but I would consider this to be a hypothesis that is actively being researched rather than something that can be asserted. In many avian taxa, structural color seems to be static as white, but maybe that's also pedantic.

Color Ordination and Comparison: I'm fond of the authors' approach of summarizing color as a multivariate trait. However, I'm not altogether certain I understand it precisely because different parts of it are described in different places throughout the manuscript. Perhaps a figure panel with a simple flow chart connecting the color and comparative analyses performed would be helpful to readers.

Lines 125-128, Figure 2B: I had some difficulty placing this analysis in the context of the study's design or tying it to any of the main hypotheses or questions. If the interpretation is that this highlights which clades tend to be the most complex, maybe a sentence about that could go in the figure legend?

Lines 213-222: I found this part of the paperless more convincing than the rest due to issues described in Uyeda et al. (2016, Syst Biol). Even with a phylogenetic PCA, we should expect biases in model fit when individual principal components are used, particularly around OU models (also see Adams and Collyer 2018, Syst Biol). So while I would interpret Figure 5B in the same way as the authors and have no reason to disagree with them, I am skeptical it allows us to really reject competition as an explanatory mechanism here.

Lines 253-254: This is really interesting and I'd love to see more written about this idea!

Lines 273-293: This is tangentially related to the previous comment, reading the discussion it felt like we dive into adaptive explanations of color without much space to discuss drift or liability/availability of different color mechanisms (these are however discussed in great detail in the introduction).

Figure 1: This is a great figure, I really enjoyed it.

Figure 3: When I see an ancestral state reconstruction like this where nearly all change is independently derived, it makes me skeptical of the model. Highly complex patterns seem to be prevalent in the Alcedininae, but I couldn't find any ancestral nodes with complexity scores half as high as extant species. Since these ancestral states are used as data in downstream analyses, it might be worth testing the robustness of those analyses to uncertainty in those states, or at least describing that uncertainty.

Figure 4: The tip colors seem to refer to a legend that is no longer in Figure 3, but in Figure 6.

[Editors’ note: further revisions were suggested prior to acceptance, as described below.]

Thank you for resubmitting your article entitled "Complex plumages spur rapid color diversification in kingfishers (Aves: Alcedinidae)" for further consideration by *eLife*.

Your revised article has been evaluated by Christian Rutz (Senior Editor) and Kaspar Delhey (Guest Reviewing Editor). The article has been improved, but there are some remaining issues that need to be addressed, as outlined below:

Ln 23 "island insularity" revise.

Ln 29-31 "Importantly, we found that island species did not have more complex plumages than their continental relatives. Thus, complexity may be a key innovation that facilitates response to relaxed (or divergent) selection pressures on islands." We cannot see how the second sentence follows the first, if island bids do not have higher plumage complexity, how can complexity be a key innovation that facilitates island living?

Ln 145 chromatic variation includes hue and saturation.

Ln 145 "achromatic color" does not make much sense; suggest referring to chromatic and achromatic variation throughout the paper rather than chromatic and achromatic color.

Ln 146 similarly "brightness" tends to be used in a multitude of ways, maybe replace with lightness or light-to-dark variation.

Ln 166-169 "To determine rates of overall plumage evolution, we used a phylogeny by McCullough et al. (2019), which incorporated thousands of ultraconserved elements (Faircloth et al., 2012) for a fully sampled, time-calibrated phylogeny of the avian order Coraciiformes (kingfishers, bee-eaters, rollers, and allies)." Revise sentence, "…we used a phylogeny…for a phylogeny…" something is off here, verb lacking.

Ln 230 maybe "theory" is too strong here, replace it with "idea"?

Ln 253 Is this the case? You mention Table 3 which refers to rates of colour evolution, not plumage complexity. This needs some clarification, as in Table 2 you do show that island species do not have more complex plumages. As a matter of fact, some sentences in this paragraph contradict the previous one.

Ln 390 some justification for the inclusion of body mass here may be good.

Ln 400 typo "drivers".

Table 3 We do have some concerns with multicollinearity here. As shown in Figure S1, different estimates of plumage complexity seem strongly intercorrelated. Thus, when your best model for chromatic variation identifies c1 and c3 as important predictors, the effects that your model quantifies constitute the effects of one predictor controlling for variation in the other. Now, if both are strongly correlated there is not that much independent variation that is relevant to explain. We think that it would be important to fit models with each predictor separately in order to check that results follow the same pattern. Moreover, this would mean reporting more complete results independent of whether the model is the "best" model identified by AIC. Potential future meta-analysts will need this information regardless of statistical significance or AIC. Please present the results from all models tested.

---

## [Author Response]

Essential revisions:1) The two main topics linked to evolutionary rates of plumage colour, island-dwelling, and plumage complexity need to be better introduced and linked in the introduction. This is a point that was made in one form or another by the three reviewers.

We have revised the introduction to make these connections more clearly. Now, we first introduce bird coloration (paragraph 1), followed by the challenges and importance of studying complexity (para 2) and a discussion of how islands are ideal systems to test evolutionary hypotheses (para 3). Finally, we move into our study system (para 4) and the hypothesis/predictions we are testing (para 5).

2) The concept of plumage complexity, which is central to the paper, requires a more comprehensive explanation. In particular, how it differs from evolutionary divergence to avoid any potential circularity.

We changed the text to clarify that complexity is an intraspecific measure, while rates are an interspecific measure (e.g., see lines 140, 170, 222, 668).

3) The methods used to analyse differences in colour evolutionary rates constitute some of the most novel aspects of the paper. However, these are difficult to follow. The authors need to cater for two different types of readers here: (a) Those that only want to broadly understand the methods used and (b) those that will want to adapt the methods to their own study system. To deal with (a) Rev 4 suggestion of a flowchart figure illustrating the methods provides an elegant way to summarise the approach used. For (b) much more detail is needed regarding the multiple ways in which analyses can vary, this ranges from information regarding arguments used within specific functions to better explanations as to why certain methodological decisions were taken.

Following these suggests, we have now added a new workflow figure (Figure 1) that we hope will clarify any questions the reader may have of our approach. We have also added several details in the Methods/Results sections to justify our methods and provide the reader with a blueprint for replicating the approach (e.g., see lines 336-359).

4) There are many statistical analyses in the paper. Some may be better suited than others (e.g. the pitfalls of using PCA suggest that maybe they are less well suited) and some seem somewhat redundant. The authors should try to streamline this as much as possible, potentially discarding ancillary analyses that do not deal with the main questions.

We removed the Bayesian models (MCMCglmm), as 2 reviewers had concerns about whether the analyses were necessary for supporting the main points of the manuscript. We further streamlined the methods as much as possible, for example, we now analyze color evolutionary rates and color convergence on islands in a unified PGLS framework (discussed in Methods, e.g., lines 377-390, 391-402).

Reviewer #1 (Recommendations for the authors):Specific comments (ln refer to line numbers)Ln 112-128 Is the PCA needed to describe colour variation? It does not seem to add much and PCA has been recently criticised in the context of comparative analyses and other multivariate approaches (some of which are implemented in the paper) are preferred (Uyeda et al., 2015; Adams and Collyer, 2018).

We felt this provided a neat way to visualize variation in complex plumage patterns across the group (as also mentioned by Reviewer 3). As such, we would prefer to leave this component in the manuscript.

Ln 139 high phylogenetic signal == species diagnostic traits. I am not sure that this should be the case, the high phylogenetic signal would indicate that closely related species have similar colours, and hence such species groups would share rather than differ in these traits. Thus, species identity among closely related species would not be easily signalled by such colours.

We changed this line to read “are more taxonomically informative than crown or wing plumage” (lines 133-134) to clarify our point.

Ln 179-180 and in passerine birds see (Cooney et al., 2022).

We appreciate the reviewer bringing this reference to our attention–we have added it at line 184.

Ln 140-187 Why did you not try to use geomorph::compare.evol.rates to compare multivariate evolutionary rates between species with low and high plumage complexity? I know this approach only deals with tips and not nodes, but it is a bit unsatisfactory to use one approach for one question and another one for the other given that the procedures are quite different.

We appreciate this comment and agree it is best to not mix-and-match methods. We were trying to account for continuous variation in plumage complexity rather than bin species into high or low values of complexity. As such, in the revised version, we treat complexity and insularity together in a unified PGLS framework, and then validate these results (as supplemental material) using compare.evol.rates with high/low complexity bins, as suggested. The results are generally in agreement with the PGLS analyses of species-specific rates, showing faster rates for more complex plumages for chromatic, but not achromatic, color.

Ln 225 why intra-specific competition?

This is just one hypothesis that could be tested in future work. We further point out that genetic drift could also explain rapid color evolution on islands (e.g., lines 103, 219, 266).

Ln 236 p=0.08 should not be interpreted as no support (see (Muff et al., 2021)), and it would be good to state the direction of this effect in the text as well (does it lean towards convergence or away from it?) as most readers will not be familiar with this particular statistic.

Point noted. We changed the text to “weak support.”

Ln 238 with 'drabber' do you mean darker (less light)? I would be careful with using terms such as drab or bright (dark and light would be better) without a clear definition first.

We changed the text to “darker” instead of drabber.

Ln 239 I am not sure that these results add much, the comparison between structural and pigmentary colours is a bit redundant (mechanisms were largely identified due to properties of the reflectance spectrum). I think that including these results detracts from the main message, similarly, I am not sure about the relevance of sex differences. Differences in mean colour between island and mainland species -if included- should come earlier in the more descriptive part of the results. Why did you not use geomorph::procD.pgls for this analysis?

We have removed this aspect of the manuscript, as 2 of 3 reviewers and the editors had concerns about it. As suggested by this reviewer, we feel it did not add very much value, and the paper is now more streamlined as a result. We did not use procD.pgls for this because, to the best of our understanding, that analysis requires an averaged value for a species, and we wanted to look at within-species variation (e.g., between sexes, among plumage patches), hence the MCMCglmm approach.

As an aside, how did you deal with sex differences in the other analyses? I might have missed it but, do they represent both males and females or only one sex? Please clarify.

In our original analysis of rates and complexity, we had averaged males and females. However, we appreciate the reviewer pointing out that we had not made this clear, or justified the approach. We now discuss this detail in the Methods (lines 370-376), provide results of a test for sexual dichromatism showing a strong correlation between male and female coloration justifying our approach of taking species averages (line 685), as well as include statistical results for males and females analyzed separately (e.g., see Supplementary files 1a-1e).

Ln 280 "… competition between individuals of the same species is driving color diversity…" this needs further explanation, what would the mechanism be?

We removed the line and reframed this section (see lines 245-270).

Ln 307-310 Why is your node+tips approach inherently better than a tips-only approach, other than by being more conservative? It could also be that including nodes increases type II error rates simply due to the inherent uncertainty associated with the estimation of ancestral traits (which often end up being an average of the tips). I think that a bit more justification is needed here.

This is a good point about error associated with ancestral states. We’ve opted to use a tip-based approach that treats species-specific rates in a PGLS framework, alongside the geomorph::compare.evol.rates approach.

Ln 312 nor achromatic variation.

We have now integrated achromatic color variation into the manuscript.

Ln 332 you averaged three spectra per patch per specimen, right?

Correct, we have added this detail at line 303.

Ln 344-46 PCA using a covariance or correlation matrix? Covariance would be the appropriate choice here as all variables are measured in the same units. This is not clear from the R code provided (at least to me). For the non-phylogenetic PCA probably the covariance matrix was used as this seems the default option for function prcomp, but I am not sure what the case is for the phylogenetic PCA function used. Nevertheless, this should be clear in the text.

We used covariance matrices, and we now indicate this in the Methods (line 332).

Ln 346 pPCA has been criticised before see (Uyeda et al., 2015; Adams and Collyer, 2018), and this is particularly relevant when choosing to discard some PCs from subsequent analyses.

We are only using PCA/pPCA to visualize trends in plumage complexity now. Both PC and pPC traits show similar trends, thus we’d prefer to leave these analyses in place. That said, we added an additional line about the caveats of pPCA (see lines 333-335).

More information on the MCMCglmm priors is needed, from the code I get the impression that default priors were used (i.e. I see no prior argument specified in the code, or am I misunderstanding something)? Such default priors are usually not good for random effects. Also, it would be good to state that models converged and how this was assessed. Also, the multivariate Wald tests used in these MCMCglmm models should be better explained in the text.

In response to comments made by the editors and Reviewer 2, we have removed this analysis from the manuscript.

The function geomorph::compare.evol.rates has two methods to establish whether evolutionary rate ratios between two regimes differ significantly from 1, the default ("simulation") and an alternative ("permutation"). In my experience using this function to analyse evolutionary rates of colour, "simulation" yields much lower p-values than "permutation" (unpubl. data). Based on reading (Adams and Collyer, 2018) and on direct communication with the main author of the geomorph package (Dean Adams) it seems that "permutation" is a better alternative to achieve an appropriate level of type I errors. Based on the R code provided it seems that the authors have used the default option "simulation". I think that the analyses using this function should be re-run with the alternative "permutation". I suspect that the evolutionary rate ratio will not differ significantly from 1 in that case. This should not matter, the result in itself is interesting regardless of the outcome, although it could mean that some key conclusions of the paper will have to be changed.

We were using geomorph 4.0.4 (which we now indicate in line 387) that has the default option set as a permutation. We clarify this in the Methods (see lines 408-410).

Figure 1. This figure could be deleted or moved to the suppl material, it does not add much to the general story.

We would prefer to leave it in place, as Reviewer 3 pointed out they liked the figure and we feel it is useful for visualizing variation in plumage complexity across kingfishers.

Figure 3C It would be good to label data points that correspond to nodes vs tips in the figure to illustrate differences in results. Would it be possible to also include island/mainland dwellings in this analysis? It should be doable to reconstruct ancestral states for this trait as well right?

We appreciate this suggestion. We have redone this figure to show both complexity (color of branches) and rates (length of branches). Our new tip-based rate analyses further incorporate both insularity and complexity (e.g., see lines 391-402).

Figure 7 I am not sure this figure should be included in the main text, it is hard to visualize the patterns.

Considering this comment and the suggestion of Reviewer 2 and the editors, we have removed this figure from the manuscript, along with the corresponding RPANDA::fit_t_comp analyses.

Reviewer #2 (Recommendations for the authors):Line 16: I think I know what you mean but this is not actually clear. By phenotypic, you mean increased rates of change per speciation event, correct? Whereas by species diversification you just mean increased speciation rates, correct? Make this clearer.

We cut this line from the manuscript.

Line 25: I'm immediately wondering how plumage complexity is calculated and whether it de facto will tend to correlate with color diversity.

We appreciate this point. To point out to the reader that these two metrics could in theory be either negatively or positively correlated, we added examples of sister kingfisher species showing simple plumages and divergent colors, as well as complex plumages with similar colors (see Figure 1B).

Lines 38-40: I'm not sure this is how the island rule is phrased. Isn't it more that organisms tend to shift notably in body size? Otherwise, how do we define a large-bodied colonist vs. a small-bodied colonist? Suggest making this sentence more general.

We removed this sentence, as it was not needed in our revised Introduction.

Line 58-59: I'm not sure what you mean here. If a color variant for a species with uniform coloration had higher fitness then yes, the whole plumage would change in tandem, but if the mutation involved a patch changing color, then the whole plumage wouldn't change in tandem. The way this is worded makes it sound like a uniformly colored species will be unable to evolve plumage complexity.

We agree this was confusing as originally written. We added text to better justify our argument (e.g., lines 57–62: “For example, in a hypothetical, uniformly colored species with strong developmental constraints that limit independent variation in color among patches, selection on the color of any single patch would cause the whole plumage to change in tandem. By contrast, if a species is variably colored (i.e., patchy, and therefore has a more complex plumage) with few constraints on the direction of color variation for different patches, selection can act on different aspects of coloration (Brooks and Couldridge, 1999).”)

Line 63: are the authors going to define plumage complexity as the number of plumage patches that are different from one another? Because, if so, this will QED correlate with color variation. Revisiting this point after a few reads of the manuscript, I realize the authors are working across intraspecific and interspecific scales in this sentence. This needs revision here.

We changed this paragraph to make our points clearer, as well as removed some redundant lines (see lines 52-68).

Lines 96-98: a lot of this feels really circular. Some attention is needed here. Cutting some words, this sentence says, "promote color divergence…contributing to high rates of color evolution and disparity". How color divergence differs from rates of color evolution differs from color disparity is not clear.

We simplified this to read “Smaller population sizes, isolation, and genetic drift could promote high rates of color evolution within island kingfishers” (lines 102-103).

Line 122: change to "sex" in Figure 2, not "individual".

Fixed.

Line 127: I'm confused. Isn't Tanysiptera monomorphic?

Generally, yes. We added a multivariate phylogenetic integration test that reveals a significant correlations between male and female plumage coloration (see lines 370-373). As such, we report the results of males and females averaged together in the main text, but we also include results for the males-only and females-only data sets in the supplement (see Supplementary files 1a-1e).

Lines 124-128: this immediately makes me wonder whether type (i) tends to be allopatric more often than does type (iii). I think the answer is yes.

This is a good question. We hope to tackle this idea in future research within the *Todiramphus* genus.

Line 137: This is in line with a recent Merwin et al. study on parrots (https://bmcecolevol.biomedcentral.com/articles/10.1186/s12862-020-1577-y).

Thank you, we have added this citation.

Lines 138-139: High phylogenetic signal in X vs Y does not mean that X is "more diagnostic of species" than Y. Re-word to clarify.

We corrected this to refer to the higher taxonomic value of characters with high versus low phylogenetic signal.

Lines 152-157: Ok, it does seem like the authors define plumage complexity as the number of uniquely colored patches.

We appreciate this comment, and we modified the text to clarify that we are counting up the “number of uniquely colored contiguous patches on the body” (see line 145).

Lines 180-187: I think there are a few key results here that deserve more space.

We have now added an additional result showing a significant effect of body size on plumage complexity and discuss this finding in line 187-189.

Line 206: the result is not that they are de novo evolving new colors, correct? Rather, they are shifting around in color space more quickly. Clarify.

We changed the text to “species distributed on islands have faster rates of chromatic color evolution.”

Lines 209-225: There are four substantial issues in this paragraph. First, anytime I hear about creating two separate trees from one to run a test I get leery. I'm not sure what the justification is here. Second, how many island species occur with a sympatric relative? I believe that the answer is not many, so we wouldn't expect interspecific competition to be relevant, correct? Third, I'm fairly sure these RPANDA functions can take a geography object to account for this varying sympatry/allopatry. No mention is made of whether this was done or, if so, how. Fourth, RPANDA isn't cited.

: Based on this and other comments made by Reviewer 1, we have opted to remove this analysis from the manuscript. However, we have gone through and made sure we cite functions at first use to credit the work of other researchers.

I may be missing something, but I also believe the last sentence here is not warranted. It could also mean that drift on islands best explains the observed pattern.

We changed the last sentence to incorporate this idea that drift on islands may explain the observed pattern.

Line 237: in what taxon?

We are not sure what the reviewer is referring to. Original line 237 stated “mainland and island species (Figure S5). This is distinct from Doutrelant et al. (2016), who showed predictable trends…” We hope that the revised paragraph has addressed this concern (e.g., see lines 195-205).

Line 247: I'm confused. Earlier you said this author DID find predictable patterns towards duller plumage. You also say it again in the discussion below.

We removed the lines about the earlier study.

Line 289: What island is this? Does New Guinea count as an island?

We do not count New Guinea as an island in our analyses because of recent and repeated connections to Australia across the Sahul Shelf during Pleistocene eustacy. The island is Halmahera Island, as we now note in the manuscript (line 100).

Lines 273-293: I am not following the logic in this paragraph. I see no reason why not finding evidence for interspecific competition influencing plumage patterns makes any statement about the relevance of intraspecific competition. This is further confused by the first discussion in the manuscript of sympatry and allopatry, which is then not actually tied into the mechanisms of interest (competition and character displacement). This is a weak point of the paper as it stands.

We changed the line about intraspecific competition and reframed the paragraph to focus on competition alongside other non-biotic factors that could explain these patterns (see lines 245-270).

Lines 300-309: I'm still a little confused about the differences between plumage complexity and evolutionary rates (it would be helpful to keep reminding the reader that one is within a species, and one is between species), but I think some of what the authors are saying here can be attributed to differences between a phylogenetic independent contrasts-esque approach (what the authors use), and a phylogenetic generalized least squares approach (what others tend to have used). The former tends to be more conservative in my experience. Per a comment above, I'm withholding too many thoughts about alternative tests until I get a clearer picture of what the authors want to test, but I suspect this could be done as a phylogenetic t-test, where the test is whether plumage complexity differs between island taxa or not.

We have reworked our analytical pipeline. This involved removing the node-based approach and replacing it with PGLS-like approaches (phylolm for univariate traits like our complexity metrics, d-PGLS for multivariate coloration).

Figure 7: very challenging to see. It might be clearer with a white background.

We removed this figure in response to comments made by R1 and the editors.

Figure S6: "Effect of data analysis on rate-complexity correlations". Data analysis is too vague. Can you come up with a more specific figure legend?

This figure has been removed.

Line 43: errant parentheses.

Fixed.

Lines 92-94: run-on sentence, passive voice. Revise.

We changed it to: “Although the family is widely distributed across the globe, their center of diversity is the Indo-Pacific, including island clades in Wallacea and Melanesia that have recently been highlighted for their high diversification rates (Andersen et al., 2018).”

Line 103: missing a word after color.

We have added “divergence” after color.

Line 291: are->is

Fixed.

Line 411: the website for this paper incorrectly lists Ian PF Owens as just PF Owens. If you check the original manuscript, it correctly lists it as IPF Owens.

Fixed.

Reviewer #3 (Recommendations for the authors):My line-by-line suggestions follow, I hope are helpful:Line 15: This is entirely pedantic, but children typically leave the cradle, whereas oceanic island lineages often remain confined there until extinction or replacement (as in Wilson's taxon cycle). Maybe there's a better analogy out there.

We appreciate the point, and this line has now been removed, following suggestions by the other reviewers on how to revise the Introduction.

Lines 57-58: Constraints on morphospace need not constrain evolutionary rates, see Goswami et al. 2014.

We appreciate this important point. We have added a more recent reference (Felice et al., 2018) near this point (line 57) that discusses the idea that rates need not be constrained by development or integration.

Line 60: Defining complexity can be a difficult topic to agree on in evolutionary biology and I think that considering the centrality of pattern complexity to the study, this would benefit from being discussed in more detail. Complexity at one level (phenotype) may not match with complexity at others (biochemistry, development, genome). I could see a working definition in the methods, as well as a supporting citation or two as a way to resolve this issue.

We added several details to the Methods (lines 337-359) and a new workflow figure illustrating the ways we are calculating plumage complexity (see Figure 1). We hope these changes have addressed this concern.

Lines 66-84: Can this paragraph be condensed and combined with the one below? I'm not seeing how it's relevant enough to the paper's key ideas to merit this much-lit review text. I don't want to hold this against the authors though, because I'm getting a subtle impression that a previous reviewer might have requested it, and nothing is more infuriating than having reviewers from different journals disagree with each other.

We condensed and combined the two paragraphs mentioned. We think it reads much cleaner now (see lines 36-51).

Line 82-84: I don't disagree with this outright – it does seem to be the case in starlings – but I would consider this to be a hypothesis that is actively being researched rather than something that can be asserted. In many avian taxa, structural color seems to be static as white, but maybe that's also pedantic.

We added a caveat (italicized) – “are considered key innovations *in some clades (e.g., African starlings; see* Maia et al., 2013b)”

Color Ordination and Comparison: I'm fond of the authors' approach of summarizing color as a multivariate trait. However, I'm not altogether certain I understand it precisely because different parts of it are described in different places throughout the manuscript. Perhaps a figure panel with a simple flow chart connecting the color and comparative analyses performed would be helpful to readers.

Great idea, we have implemented it in a new Figure 1.

Lines 125-128, Figure 2B: I had some difficulty placing this analysis in the context of the study's design or tying it to any of the main hypotheses or questions. If the interpretation is that this highlights which clades tend to be the most complex, maybe a sentence about that could go in the figure legend?

Great idea, we added the line “Clades with more complex plumages tend to have a higher proportion of among-patch variation” to the legend (lines 685-686).

Lines 213-222: I found this part of the paperless more convincing than the rest due to issues described in Uyeda et al. (2016, Syst Biol). Even with a phylogenetic PCA, we should expect biases in model fit when individual principal components are used, particularly around OU models (also see Adams and Collyer 2018, Syst Biol). So while I would interpret Figure 5B in the same way as the authors and have no reason to disagree with them, I am skeptical it allows us to really reject competition as an explanatory mechanism here.

We have opted to address competition in an alternative way (e.g., including the number of sympatric species as a predictor of color rate variation in our PGLS models).

Lines 253-254: This is really interesting and I'd love to see more written about this idea!

Thank you for the encouraging remark.

Lines 273-293: This is tangentially related to the previous comment, reading the discussion it felt like we dive into adaptive explanations of color without much space to discuss drift or liability/availability of different color mechanisms (these are however discussed in great detail in the introduction).

Good point. We have now condensed much of the introduction (see lines 36–51) and added further discussion about color mechanisms and rates of evolution (e.g., see lines 256-270: “Another mechanism that could explain the observed rapid color evolution on islands is divergence in abiotic factors among islands…”).

Figure 1: This is a great figure, I really enjoyed it.

We appreciate this positive comment about this figure.

Figure 3: When I see an ancestral state reconstruction like this where nearly all change is independently derived, it makes me skeptical of the model. Highly complex patterns seem to be prevalent in the Alcedininae, but I couldn't find any ancestral nodes with complexity scores half as high as extant species. Since these ancestral states are used as data in downstream analyses, it might be worth testing the robustness of those analyses to uncertainty in those states, or at least describing that uncertainty.

Thank you very much for pointing this out. We realized there was an error in that we were showing ancestral estimates derived from a scaled tree, but the tree shown was unscaled, thus making the ancestral states appear “off”. We would like to point out that our analyses were correct, it was just an issue with visualization. The issue has been corrected in the new figure (see Figure 4).

Figure 4: The tip colors seem to refer to a legend that is no longer in Figure 3, but in Figure 6.

We fixed this in the revised version (old Figure 3 was merged with Figure 4, and the island legend added; see new Figure 4).

[Editors’ note: further revisions were suggested prior to acceptance, as described below.]

Ln 23 "island insularity" revise.

We removed "island"

Ln 29-31 "Importantly, we found that island species did not have more complex plumages than their continental relatives. Thus, complexity may be a key innovation that facilitates response to relaxed (or divergent) selection pressures on islands." We cannot see how the second sentence follows the first, if island bids do not have higher plumage complexity, how can complexity be a key innovation that facilitates island living?

We rephrased to:

"Thus, complexity may be a key innovation that facilitates evolutionary response of individual color patches to relaxed (or divergent) selection pressures on islands rather than being a direct target of selection itself" (see L29-32).

We hope this makes our point more clear–that complexity is not itself changing on islands but enables change of the color of individual patches under drift or selection on different islands.

Ln 145 chromatic variation includes hue and saturation.

Noted and added this distinction where applicable (e.g., L144, 660, 671).

Ln 145 "achromatic color" does not make much sense; suggest referring to chromatic and achromatic variation throughout the paper rather than chromatic and achromatic color.

We appreciate this comment and revised all instances throughout (e.g., to "achromatic variables" in lines 350, 405, 671, or "achromatic variation" in lines 157, 726).

Ln 146 similarly "brightness" tends to be used in a multitude of ways, maybe replace with lightness or light-to-dark variation.

We implemented the suggested wording throughout the manuscript.

Ln 166-169 "To determine rates of overall plumage evolution, we used a phylogeny by McCullough et al. (2019), which incorporated thousands of ultraconserved elements (Faircloth et al., 2012) for a fully sampled, time-calibrated phylogeny of the avian order Coraciiformes (kingfishers, bee-eaters, rollers, and allies)." Revise sentence, "…we used a phylogeny…for a phylogeny…" something is off here, verb lacking.

We changed the wording to "To determine rates of overall plumage evolution, we used a recent time-calibrated phylogeny (McCullough et al. 2019) that incorporated thousands of ultraconserved elements (Faircloth et al., 2012) and fully sampled the avian order Coraciiformes (kingfishers, bee-eaters, rollers, and allies)." We hope the editors agree this is more clear.

Ln 230 maybe "theory" is too strong here, replace it with "idea"?

We changed "theory" to "idea" as suggested (see L229).

Ln 253 Is this the case? You mention Table 3 which refers to rates of colour evolution, not plumage complexity. This needs some clarification, as in Table 2 you do show that island species do not have more complex plumages. As a matter of fact, some sentences in this paragraph contradict the previous one.

We carefully re-read this paragraph and reworked it to make our points more clear. For example, we moved the line about lizard dewlap diversity up to where we discuss our result of increased plumage diversity on islands (L251), added clarification (e.g., "low amounts of plumage variation" in L256), and removed redundant lines (e.g., a line in the middle of the paragraph starting with "However, we found the opposite…" was redundant with current L250).

Ln 390 some justification for the inclusion of body mass here may be good.

We added justification to L386:

"Body mass has recently been shown to explain variation plumage complexity of passerine birds (Cooney et al., 2022), therefore we also included ln body mass (in grams) as a covariate in our regression models (species averages obtained from Dunning, 2007)."

Ln 400 typo "drivers".

We fixed the typo.

Table 3 We do have some concerns with multicollinearity here. As shown in Figure S1, different estimates of plumage complexity seem strongly intercorrelated. Thus, when your best model for chromatic variation identifies c1 and c3 as important predictors, the effects that your model quantifies constitute the effects of one predictor controlling for variation in the other. Now, if both are strongly correlated there is not that much independent variation that is relevant to explain. We think that it would be important to fit models with each predictor separately in order to check that results follow the same pattern. Moreover, this would mean reporting more complete results independent of whether the model is the "best" model identified by AIC. Potential future meta-analysts will need this information regardless of statistical significance or AIC. Please present the results from all models tested.

We appreciate this comment. We now calculate variance inflation factors (which we not are mostly rather low, which a max of ~4) and present results after removing each complexity metric in turn. As suggested, we also include the full regression results. We describe our approach in Methods L409-415 and included new Figure 3—figure supplement 1 (showing the "full" model results and sub-models with complexity metrics dropped in turn) and Supplementary files 1f and 1g (results for all models tested). Effect sizes were ~stable under these alternative model formulations (Figure 3—figure supplement 1), suggesting our results are robust to the observed levels of multicollinearity.

We would like to point out that the new Supplementary files 1f and 1g are rather unwieldy (63 models tested in each table). As such, we would ask the editor(s) to consider whether the new Figure 3—figure supplement 1 is sufficient for addressing this important point about potential multicollinearity influencing our results.